



# A conceptual model of organochlorine fate from a combined
# analysis of spatial and mid/long-term trends of surface and
# ground water contamination in tropical areas (FWI)
Philippe CATTAN[1,2], Jean-Baptiste CHARLIER[3], Florence CLOSTRE[4,5], Philippe
LETOURMY[6,7], Luc ARNAUD[8], Julie GRESSER[9], Magalie JANNOYER[10]
[1] CIRAD, UPR GECO, Montpellier, F-34398, France.
[2] GECO, Univ Montpellier, CIRAD, Montpellier, France.
[3] BRGM, Univ. Montpellier, Montpellier, F-34000, France
[4] CIRAD, UPR HortSys, Le Lamentin, F-97285, Martinique, France.
[5] HortSys, Univ Montpellier, CIRAD, Montpellier, France.
[6] CIRAD, UPR AIDA, Montpellier, F-34398, France.
[7] AIDA, Univ Montpellier, CIRAD, Montpellier, France.
[8] BRGM, Fort-de-France, F-97200, Martinique, France
[9] ODE, Fort-de-France, F-97201, Martinique, France
[10] CIRAD, DGDRS, Montpellier, F-34398, France.
*Correspondence to*: Philippe Cattan (philippe.cattan@cirad.fr)
**Absract.** In this study, we investigated the management of long-term environmental pollution by organic
pollutants such as organochlorine pesticides. We set out to identify conditions that are conducive to reducing
pollution levels for these persistent molecules and then propose a conceptual model of organochlorine fate in
water. Our approach looked at spatio-temporal changes in pollutant contents in surface water (SW) and
groundwater (GW) on a large scale, in order to decipher the respective roles of soil, geology, hydrology and past
treatment practices. The case of chlordecone (CLD) on the island of Martinique (1,100 km²) was selected given
the sampling campaigns carried out since 2007 over more than 150 sites. CLD, its metabolite chlordecone-5b-
hydro (5bCLD) and the metabolite/parent compound ratio were compared. As regards the spatial variability of
water contamination, our results showed that banana cropping areas explained the location of contaminated SW
and GW, whereas the combination of soil and geology factors explained the main spatial variability in the
5bCLD / CLD ratio. For temporal variability, these conditions defined a high diversity of situations in terms of
the duration of pollution, highlighting two groups: water draining old geology & ferralsols or vertisols *vs.* recent
geology & andosols. A theoretical leaching model provided some key information to help interpret downward
trends in CLD and 5bCLD observed in water. Lastly, a conceptual model of organochlorine fate is proposed to
explain the diversity of the 5bCLD/CLD ratio in water. Our conclusions highlight the combined role of soil and
groundwater residence time for differentiating between conditions that are more conducive, or not, to the
disappearance of CLD from the environment. This paper presents a model that provides an overall perception of
organochlorine pesticide fate in the environment.

**Keywords**
Pesticide; Surface water; Groundwater; Temporal dynamics; Chlordecone





## 1 Introduction

The pollution of rivers and aquifers by persistent organic pollutants (POPs) and organochlorine pesticides is a global issue (Gonzalez et al., 2012; Masih et al., 2014; Montuori et al., 2014; Zhang et al., 2004). Their long-term persistence after application (i.e. several decades to several centuries) raises the question of what is polluted and to what level, and how to manage and live with pollution. Moreover, the environment is not uniformly contaminated. Interactions between human pesticide application practices and environmental conditions lead to high variability in the contamination level of environmental compartments. This variability can be perceived by observing surface water (SW) and groundwater (GW) contamination.

Globally, changes in pesticide applications over several decades have resulted in downward and upward trends for pesticide concentrations in SW (Ryberg and Gilliom, 2015; Stone et al., 2014). This is also the case for GW, for which contamination trends have illustrated the leaching of pesticides from soils towards aquifers on a regional scale (Bexfield, 2008; Kolpin et al., 2004; Lapworth et al., 2006). Quality in SW is highly correlated to that in GW, due to strong interactions between aquifers and rivers on a catchment scale. Surprisingly, there is a lack of studies combining both SW and GW observations in order to characterize pollution in all the compartments (shallow and deep) of the hydrological cycle. Thus, this article addresses the issue of the conditions and processes that determine the spatial distribution of a persistent pollutant in water on a regional scale, investigating the case of chlordecone contamination in the French West Indies.

Chlordecone (CLD, $C_{10}Cl_{10}O$; CAS number 143-50-0; 491 g mol$^{-1}$) is an organochlorine classified as a POP (U. S. Environmental Protection Agency, 2012; UNEP, 2007). Numerous issues stem from CLD use in the French West Indies (FWI) (Lesueur Jannoyer et al., 2017). CLD was used from 1970 to 1993 to control the black weevil (*Cosmopolites sordidus*) in banana plantations. Application intensity greatly depended on the farmers (Cabidoche et al., 2009; Della Rossa et al., 2017; Levillain et al., 2012) and introduced high spatial variability in soil contamination. Despite its worldwide ban in 1992, CLD continues to contaminate aquatic ecosystems in different parts of the world (Coat et al., 2011; Luellen et al., 2006). As a consequence, CLD-polluted soils in FWI go on to contaminate GW (Arnaud et al., 2017; Gourcy et al., 2009) and rivers (Bocquene and Franco, 2005; Coat et al., 2011; Crabit et al., 2016; Mottes et al., 2015; Observatoire de l'Eau de la Martinique et al., 2012). This pollution raises concerns, as CLD causes adverse effects on health, both from acute and environmental exposure (Cannon et al., 1978; Cordier et al., 2017; Multigner et al., 2015).

The persistence of pesticides in soils and their transfer to percolation water depend on various processes, such as degradation and sorption, influenced by molecule properties, and the soil and climate context (Arias-Estévez et al., 2008). For CLD, adsorption on soil aggregates, hence the risk of water pollution, greatly depends on soil type, as indicated by the partitioning coefficient (Koc) between the sorbed part on soil organic matter, which varies from 2.5 to 20 m$^3$ kg$^{-1}$ (Cabidoche et al., 2009; Woignier et al., 2012). Moreover, at depth, contrasting residence times in aquifers of several years to several decades partly account for the variability in GW contamination by CLD (Gourcy et al., 2009).

Recent studies highlighted the fact that degradation can occur for this molecule (Fernández-Bayo et al., 2013; Mouvet et al., 2017). CLD-5b-hydro (5bCLD, $C_{10}Cl_9HO$; CAS number 53308-47-7; 456 g mol$^{-1}$) is a mono-hydrochlordecone, which can be produced as an impurity during CLD manufacturing (Cabidoche et al., 2009;



Fernández-Bayo et al., 2013). It has also been obtained experimentally by degradation of CLD through
photolysis and microbial degradation (Orndorff and Colwell, 1980; Wilson and Zehr, 1979). Orndorff and
Colwell (1980) interpreted the in situ value of 5bCLD content as an indicator of the degradation process.
Studying both the fate of the parent and metabolite compounds, or their ratio, provides a more complete
understanding of the transportation of the molecule (Farlin et al., 2017; Gassmann et al., 2013; Kolpin et al.,
2004). 5bCLD has been found in soils, waters and food webs, along with CLD but at much lower levels (Borsetti
and Roach, 1978; Clostre et al., 2015; Coat et al., 2011; Devault et al., 2016; Observatoire de l'Eau de la
Martinique et al., 2012).
To sum up, in FWI, human practices and the physical environment lead to high variability conditions for CLD
and 5bCLD that may impact the environment. Our aim was to identify the conditions that are conducive to a
decrease in pollution levels, in order to propose a conceptual model of organochlorine fate in water. We focus
here on river contamination, which is driven by all the environmental compartments, being consequently an
integrative survey site of land-use, soil variability, and aquifer contributions. Based on the sampling campaigns
in Martinique (FWI) since 2007, we explore river contamination trends over time and the relationships between
surface and underground CLD rates in water. Spatial and temporal distributions of contamination are interpreted
according to soil and geology mapping, hydrology and past CLD treatment practices. This work will lead on to
identifying areas with a low or high impact on water pollution, in order to manage polluted areas more
effectively.

## 2 Material and methods

### 2.1 Study site

*Location and climate.* The study area covered the volcanic island of Martinique (1,100 km²) in the French West
Indies in the Caribbean (Figure 1). The climate is tropical, hot and humid. Annual rainfall is almost a linear
function of altitude (0 to 1,500 m ASL) and ranges from 2,500 to 10,000 mm on the east coast, and 1,000 to
10,000 mm on the west coast.
*Geology*. Eight volcanic units (grouped into 3 simplified types according to the age of the volcanic arcs) have
been identified (Germa et al., 2010, 2011) – see Figure 3 for an overview of the geological map: (1) Basal
Complex and Sainte Anne Series (24.8±0.4–20.8±0.4 Ma) for the older arc; (2) Vauclin–Pitault Chain
(16.1±0.2–8.44±0.12 Ma) and (3) South-western Volcanism (9.18±0.16–7.10±0.10 Ma) for the intermediate arc;
and (4) Morne Jacob volcano (5.14±0.07–1.54±0.03 Ma), (5) Trois Ilets Volcanism (2.358±0.034 Ma and
346±27 ka), (6) Carbet Complex (998±14 to 322±6 ka), (7) Mount Conil (543±8 to 127±2 ka) and (8) Mount
Pelée (126±2 ka to present) for the recent arc. The volcanism is andesitic with predominantly explosive
volcanoes. Geological formations are thus composed by ash flows, lava flows, and reworked formations (e.g.
lahars and debris flows) channelled in peripheral valleys, and atmospheric fallout on a larger scale. Such geology
generates a high spatial variability of lithology strata and contrasting weathering levels between geological units.
*Soils.* Two climate sequences of soils (IUSS Working Group WRB, 2014) are found in Martinique according to
Colmet-Daage et al. (Colmet-Daage et al., 1965) – see Figure 3 for an overview of the soil map: (1) ferralitic



soils (latosols) -> ferralsols -> vertisols and (2) andosols -> nitisols. All primary minerals of andesitic rocks are
weathered, so that soils have a high content of secondary minerals: halloysite for nitisols, halloysite and Fe-
oxihydroxides for ferralsols, and allophane for andosols. In addition, Martinique has skeletal andosols and young
raw soil containing pumice gravels, deriving from recent pyroclasts. All these soil types are acidic. Carbon
contents are unusually high for tropical soils, in particular for untilled andosols, and range from 10 to 140 g kg$^{-1}$
according to Cabidoche et al. (2009) and Brunet et al. (Brunet et al., 2009). These features may induce large
differences in pesticide fate in soils. Since the soil types "poorly developed soil on ash and pumice" and
"andosol" are similar and rich in allophanes, in this study they have been grouped under the designation
'andosol'. Likewise, fersiallitic soils and ferralsols have been grouped under the designation 'ferralsol' (very
dominant among the two soil types), as they are both rich in kaolinites and they are intergrades resulting from
the alteration of ferralitic soils (Colmet-Daage et al., 1965; Quantin et al., 1991).
***Hydrology, hydrogeology and contamination.***  High rainfall intensities during tropical storms generate flash
floods with a torrential regime in the rivers of Martinique. Permeable soils in the Lesser Antilles favour
infiltration and aquifer recharge (Charlier et al., 2008). As a consequence, hydrological studies on a catchment
scale showed that the water budget on an annual scale is mainly controlled by underground processes, limiting
surface runoff contributions (Charlier et al., 2008, 2011). Stream flows are greatly influenced by SW/GW
interactions, leading to consider that GW drainage is a major process of river contamination (Arnaud et al., 2017;
Charlier et al., 2009; Morgenstern et al., 2015; Mottes et al., 2015). At depth, most of the volcanic aquifers are
small, a few km² at most, as a result of the complex geological structure, which has undergone several phases of
volcanism, erosion and weathering (Lachassagne et al., 2014; Vittecoq et al., 2015). As shown by Charlier et al.
(Charlier et al., 2015), who compared the hydrogeological functioning of aquifers with contrasting lithologies
and age formations, the groundwater residence time is highly variable, between a few years for recent
unweathered formations, to several decades for old weathered formations. It may result in various levels of river
contamination by CLD linked to the hydrogeological context of the catchment.
**2.2 Building up the Database**
**2.2.1 CLD and 5bCLD sampling in water**
The study period ran from the end of 2009/early 2010 to 2014. Since 2009-2010, 5bCLD has been analysed on a
routine basis with CLD. For SW, we used data from a programme monitoring water quality carried out by the
Martinique Water Office throughout Martinique and from a research programme implemented by CIRAD
(CIRAD, F-97285 Le Lamentin, Martinique, France) in the Galion watershed in Martinique. Sampling was
carried out manually according to standard NF EN ISO 5667-3 and the FD T 90-523-1 guideline. For GW, we
used data from a programme monitoring groundwater quality carried out by BRGM throughout Martinique. The
sampling methodology was based on standard NF EN ISO 5667-3, and the FD T 90-523-3 and FD X31-615
guidelines. Before sampling in wells, at least three purge volumes were pumped with a submersible pump until
stabilization of the chemical groundwater parameters. Samples were stored at 5°C and shipped in ice coolers to
the BRGM analytical laboratory in Orléans, France.



### 2.2.2 CLD and 5bCLD analysis

5bCLD is the main CLD co- and alteration product of CLD for which a commercial analytical standard is available. Reference standards for CLD and 5bCLD were purchased from Dr. Ehrenstorfer GmbH (Augsburg, Germany) for both laboratories.

For SW, samples were analysed at the LDA26 laboratory. CLD and 5bCLD sample analyses were carried out by liquid/liquid extraction (Dichloromethane and ethyl acetate 80/20) followed by Ultra–High-Performance liquid chromatographic separation and mass spectrometric identification. An Ultra–High-Performance liquid chromatography tandem mass spectrometry analysis was performed with a Thermo electron system (TSQ Quantum Ultra) or ABSciex system (API4000 or API4000 Q-Trap). The compounds were separated on an Alltima C18 (5µm-150 x 2.1mm). Two transitions were monitored $506.7 > 426.5$ and $506.7 > 424.5$ for CLD and $472.6 > 392$ and $472.6 > 454.5$ for 5bCLD. 2.4D d3 was used as the internal standard for calibration. The key parameters of the method (linearity, repeatability, interday precision, specificity, extraction efficiency and limit of quantification) were validated in accordance with the standard NF T 90-210 method (AFNOR 2009). The CLD and 5bCLD limits of quantification were determined by spiking natural surface water samples.

For GW, samples were analysed at the BRGM laboratory in Orléans, France. A gas chromatography tandem mass spectrometry analysis was carried out with a Bruker system (Marne la Vallée, France) composed of a GC450 gas chromatography apparatus equipped with an 1177 injector, a Combi Pal (CTC) autosampler and a 300MS triple quadrupole mass spectrometer. The injector was equipped with a $4 \times 6.3 \times 78.5$ mm liner with fibreglass and Sky™ deactivation. The compounds were separated on an Rxi-1MS (30 m, 0.25 mm ID, 0.25 µm) column from Restek (Lisses, France). CLD and 5bCLD analyses of water samples were carried out by liquid/liquid extraction followed by gas chromatographic separation and mass spectrometric identification. The key parameters of the method (linearity, repeatability, interday precision, specificity, extraction efficiency and limit of quantification) were validated in accordance with the standard NF T 90-210 method (AFNOR, 2009). The CLD and 5bCLD limit of quantification were determined by spiking natural water samples.

Both the LDA26 and BRGM laboratories are accredited for pesticide analysis and are involved in proficiency testing schemes organized by ANSES (French Agency for Food, Environmental, and Occupational Health and Safety), thereby ensuring the quality and coherence of the results. The limits of CLD and 5bCLD quantification in water were different for LDA26 and BRGM: 0.01 and 0.03 µg.L$^{-1}$, respectively. By convention, the limits of detection were set at one third of the limits of quantification, i.e. 0.003 and 0.01 µg.L$^{-1}$ for LDA26 and BRGM, respectively.

### 2.2.3 Value assessment and factors

*Value assessment.* For calculation, a value of 10% of the quantification limit was assigned when the compound was not detected (i.e. 0.001 for LDA26 or 0.003 µg L$^{-1}$ for BRGM), and an intermediate value of 0.006 µg.L$^{-1}$ was assigned when the compound was detected but not measurable at LDA26.

*Factors.* The statistical analysis set out to assess the effect of various environmental factors - soils, geology, hydrological sectors, historical banana area, and time - on CLD and 5bCLD concentrations and on the 5bCLD / CLD ratio, determined at each sampling point. For the soil factor, as the water at one sampling site originated



from a watershed possibly draining various soil types, we associated each sampling point with the main soil type
of the watershed drained by the sampling point according to the soil map of Colmet-Daage et al. (1965). For the
other factors, each sampling point was associated with the factor value at the sampling point.

**2.3 Selection of data and statistical analysis**

**2.3.1 Range of contamination values**

The relevance of contamination was assessed according to the EU 'Water Framework' and 'Quality of drinking
water' Directives (European Union, 1998, 2000) and their transposition into French law (French government,
2001). Three thresholds of water contamination classes stemmed from these directives: 0.1, 0.5 and 2.0 $\mu g L^{-1}$.
The first two regulatory thresholds apply to the mean annual content in tap water intended for human
consumption: 0.1 $\mu g L^{-1}$ is the threshold for each pesticide (threshold applying to CLD), and 0.5 $\mu g L^{-1}$ is the
threshold for the sum of all pesticides. Raw water exceeding these thresholds needs to be treated for human
consumption. The third value, 2.0 $\mu g L^{-1}$ is the threshold beyond which, according to the regulation, water can no
longer be termed drinkable even after treatment. The threshold values of 0.1 and 0.5 $\mu g L^{-1}$ are also chosen to
define good environmental status.

**2.3.2 Data selection**

*Global data set.* For SW, the data set consisted of 1,866 analyses from 136 sampling points and 76 rivers. The
analyses were not evenly distributed. Most of the sampling points had a low measurement frequency (105 had
fewer than 5 analyses) and only 18 sampling points had more than 50 analyses covering the entire 2009-2014
period. However, the number of analyses per complete year varied between 188 and 352.   For GW, the data set
consisted of 282 analyses from 21 sampling points and 6 water bodies. Basically, sampling occurred twice a year
at each sampling point. At three sampling points, sampling occurred monthly in some years.
*Data selection for statistical models.* For statistical analysis, we discarded data where CLD concentrations were
below detection limits (and consequently 5bCLD concentrations too, as 5bCLD concentrations are always lower
than CLD), as they would have led to an inappropriate ratio value (ratio of 1 according to the value assessment
rule described below). Additionally, although we gathered data from contaminated areas, some of the water
samples were contaminated with CLD, but no 5bCLD was detectable. For the statistical analysis, we kept all the
data (with and without measurable 5bCLD) from sampling points for which at least half the samples had
measurable 5bCLD contents ($\geq$ 0.03 or 0.01 $\mu g L^{-1}$). This avoided overestimating the concentration for the
sampling point, which would have been the case if we had discarded all the data with no measurable 5bCLD. For
SW, we selected 963 data items. This SW data set covered 38 sampling points out of a total of 136. For GW, we
selected 123 data items. This GW data set came from 7 sampling points.
*Data selection for temporal analysis on specific rivers.* In order to highlight differences between pesticide
trends depending on the sampling point, we chose rivers for which the analysis covered the entire 2009-2014
period. This led to the selection of 14 sampling points, all having more than 50 analyses. As stated above, we
discarded analyses where CLD and 5bCLD contents were below detection limits.



### 2.3.3 Statistical analysis

*Models*. To ensure that the residue distribution of the analysis of variance (ANOVA) model followed the assumptions of equal variance and normality, we used log transformed (natural log) data. We analysed our SW and GW data sets by a multi-way analysis of variance using the MIXED procedure in SAS software (SAS Institute Inc, 2002). The effects to be taken into account in the models were chosen by comparison of the AIC (Akaike Information Criterion).

Model 1 was used on the SW data set to test different effects on the CLD content, the 5bCLD content and the ratio of the 5bCLD content to the CLD content in SW. The soil and geology factors were dependent on each other. For this reason, only combinations of these 2 factors were considered in the model.

Eq. (1) $$Y_{ijklm} = \mu + \alpha_i + \beta_{ij} + \gamma_t + D_{ijk} + \varepsilon_{ijtkl}$$

where $Y_{ijklm}$ is the observation (i.e. ln(5bCLD), ln(CLD) or ln(5bCLD /CLD) ), $\mu$ is the general mean, $\alpha_i$ is the (soil x geology) type effect, $\beta_{ij}$ the hydrological sector effect for each (soil x geology) type, $\gamma_t$ is the date effect, $D_{ijk}$ the random effect of the sampling point for each (soil x geology) type and $\varepsilon_{ijtkl}$ is the residual error.

Model 2 was used on the GW data set. Both soil and geological factors were totally correlated for the GW data set, making it impossible to distinguish the soil effect from geology; likewise for groundwater basins and hydrographic sectors. Consequently, only soil and hydrogaphic sectors were tested for model 2:

Eq. (2) $$Y'_{ijklm} = \mu' + \alpha'_i + \beta'_{ij} + \gamma'_t + D'_{ijk} + \varepsilon'_{ijtkl}$$

where $Y'_{ijklm}$ is the observation (i.e. ln(5bCLD), ln(CLD) or ln(5bCLD /CLD) ), $\mu'$ is the general mean, $\alpha'_i$ is the soil type effect, $\beta'_{ij}$ the hydrogaphic sector effect for each soil, $\gamma'_t$ the date effect, $D'_{ijk}$ the random effect of the sampling point for each soil and $\varepsilon'_{ijtkl}$ is the residual error.

As our data set was log transformed, dispersion indices were calculated as half the difference between the limits of the confidence interval (confidence coefficient: 0.68).

The significance of the sampling point effect was assessed by comparison of - 2 log-likelihood from the models with and without the sampling point as the random effect, as this difference followed a chi-square distribution under the null hypothesis.

*Trend analysis.* For SW, to study temporal trends, we selected estimated means of the time series for each date. Autocorrelations were assessed with the Durbin-Watson test and monotonic trends were assessed with the Mann-Kendall (MK) test. We calculated Sen trends for each variable (CLD, 5bCLD and ratio) in order to compare dynamics for the two compounds.

### 2.3.4 Conceptual model of CLD fate

A simple iterative leaching model was developed to assess the evolution of CLD, 5bCLD and the 5bCLD / CLD ratio over time. This model expressed that the 5bCLD / CLD ratio in water equally depended on degradation and transfer rates as well as the remaining storage of CLD and 5bCLD in soils. The governing equations are given below:

Eq. (3): CLD storage in soil $$CLD(t + 1) = CLD(t) - CLD(t) \times T_{CLD} - CLD(t) \times C_{degrad}$$

Eq. (4): 5bCLD storage in soil $$5bCLD(t + 1) = 5bCLD(t) - 5bCLD(t) \times T_{5bCLD} + CLD(t) \times C_{degrad}$$

Eq. (5): ratio in water $$5bCLD/CLD = (CLD(t) \times T_{CLD})/(5bCLD(t) \times T_{5bCLD})$$





$T_{CLD}$ and $T_{5bCLD}$ are the rates of lixiviation for CLD and 5bCLD, respectively, $C_{degrad}$ the rate of CLD degradation
into 5bCLD, and t the time. CLD and 5bCLD are expressed in units of mass. According to data reported by
Cabidoche et al. (2009), considering an area of 1 m² and that pollutants are distributed within the first 0.3 m of
soil, $T_{CLD}$ is expressed as follows:
Eq. (6): $$T_{CLD} = \frac{R}{K_{oc} \times (C/1000) \times D \times (0.3 \times 1000)}$$
where Koc (L kg$^{-1}$) is the partitioning coefficient between the sorbed part on soil organic matter and the
dissolved part in water, D (kg dm$^{-3}$) the bulk density, C (g kg$^{-1}$) the soil carbon content, and R (mm) the annual
amount of rainfall.
The calculation steps are given below:
-    the initial CLD and 5bCLD stocks are set to 100 and 0 units of mass respectively
-    calculation of leached CLD quantities (Eq. (3))
-    calculation of degraded CLD quantities, i.e. transformed in 5bCLD (Eq. (3))
-    calculation of remaining CLD quantities (Eq. (3))
-    calculation of leached 5bCLD quantities (Eq. (4))
-    calculation of remaining 5bCLD quantities (Eq. (4))
-    calculation of mass ratio in water (Eq. (5)) that accounts for concentration ratio since the two

274        compounds are leached with the same water quantities.

**3 Results**
**3.1 Variability of CLD contamination and its relationships with 5bCLD**
Figure 2 shows the relationship between the means of 5bCLD and CLD in rivers at each sampling point. We
found that the water 5bCLD content was at least tenfold lower than the water CLD content. However, there was
not a unique relationship between 5bCLD and CLD. The frequency distribution of the means of the 5bCLD to
CLD ratio in SW and GW clearly showed that a threshold of 0.07 divided the data set into two groups: a low and
a high ratio around 0.02 and 0.1, respectively.
**3.2 Spatial analysis**
**3.2.1 General distribution**
Figure 3 presents the CLD concentrations (top) and the 5bCLD/CLD ratio (bottom) for SW (square/triangle) and
GW (star) throughout Martinique, according to hydrological sectors (left), soil (middle), and geology. The top of
Figure 3 shows that the most challenging areas relative to CLD contamination were mainly situated in the
northern Atlantic and central part of Martinique. The distribution for the 5bCLD / CLD ratio was different. The
bottom of Figure 3 shows that the group with the high ratio (>0.07) was mainly located either in the highly
contaminated northern areas, or in some parts of the low-contamination areas in southern and western
Martinique.



We observed overall consistency between the distribution of SW and GW contamination: the higher the CLD content or 5bCLD / CLD ratio for SW, the higher the CLD content or 5bCLD / CLD ratio for GW. However, the west coast displayed some exceptions, since we observed contaminated GW (low contamination most of all) while CLD was not detected in the rivers in the neighbourhood. Similarly, the 5bCLD / CLD ratio for GW belonged to the high value group (>0.07 µg L$^{-1}$), while the 5bCLD / CLD ratio for SW belonged to the low value group, or was not available because of no contamination.

### 3.2.2 Impact of physical conditions

*Land-use practices: high level of contamination in historical banana areas*

Globally, for the water CLD content, the SW and GW contamination sites matched with the historical banana areas since 1970, i.e. during CLD application. Surprisingly, SW and GW contamination occurred outside these banana areas. This was mostly with low concentrations under 0.1 µg l$^{-1}$ and rarely with the higher levels (one point in the South-West for GW, far from the banana area,). Most of these isolated points had a high 5bCLD / CLD ratio, leading the 5bCLD / CLD ratio not to match the banana field distribution, suggesting past CLD misuse.

*Hydrographic sector: a functional relationship between measurement points*

Introducing hydrographic subsectors made it possible to establish a functional relationship between measurement point data. Notably, this helped to explain why some points close to each other did not have the same contamination level. For example, the CLD content of hydrographic subsector 1 (see Figure 3 left for locations) was different from hydrographic subsector 2 even though the points in each zone had the same contamination level. However, some differences could occur on the north-east coast. This was encountered in zone 3, where the contamination levels seemed to be linked to the altitudinal gradient. Contamination increased downwards in coherence with the banana field distribution along the coast at the lowest altitudes. The statistical results summarized in Table 1 confirm this interaction between hydrographic sectors and soil /geology for CLD and GW. However, no effect was found for the 5bCLD content and the 5bCLD / CLD ratio.

*Soil type: a factor explaining some ratio variations in SW*

Table 1 shows significant differences in GW CLD contamination according to the soil/geology pair: GW on nitisols, which are associated with old formations (older than 1 My), was more contaminated than on andosols associated with recent formations (1My to present). This did not result in any significant difference for SW. However, for SW, we observed significant differences for the 5bCLD / CLD ratio, opposing a low ratio for nitisols to a higher ratio for andosols (Figure 4). We also noted a higher ratio for vertisols. This statistically confirmed the result mapped in Figure 3, showing high 5bCLD / CLD ratios on vertisols in southern Martinique.

*Geology: a factor explaining ratio variations in SW and GW*

The age of the main geological units was used as an indicator of hydrogeology and notably residence time in the aquifers, which is linked to pesticide transfer kinetics in GW, as well as in SW fed by it. Thereby, shorter residence times were observed for more recent formations. It can be seen in Figure 3 that the highest CLD contents in water matched with recent geological formations in the banana cropping area (northern half of the island). Medium and low CLD contents were observed in other older geological units, or outside banana





cropping areas. As regards the 5bCLD / CLD ratio, the highest values were only observed in the most recent
units (0.5 My to present), for the most contaminated water bodies in the North Atlantic area (not shown).

It is interesting to note that the soil effect depended on geology. Figure 4 illustrates this, presenting the mean
ratio for each soil type according to the age of the geological formations. For andosols and ferralsols/andosols,
the ratio appeared to be significantly higher for recent geology.

To sum up, banana cropping areas explained the location of contaminated SW and GW, whereas the
combination of soil and geology factors explained the main spatial variability of the 5bCLD / CLD ratio, with
the highest values in the North associated with recent geological units and the highest values in the South
associated with vertisols.

### 3.3 Temporal analysis

#### 3.3.1 Pesticides evolve differently in GW

Figure 5 illustrates pesticide trends in GW for the three longest available time series. The mean CLD content
globally decreased for two sites (Chalvet and Source Morne Figue) and remained stable for Lelene, while the
5bCLD content had a more erratic evolution, probably due to the greater influence of hydrological conditions
(climatic seasonality). As pointed out by Arnaud et al. (2016), these contamination periods correspond to rising
and falling groundwater levels, and therefore to periods of aquifer recharge. For the two sites showing a decrease
in water CLD content, water 5bCLD contents below the detection limit appeared less frequently, and completely
disappeared in the case of the Source Morne Figue site after 2011. This was consistent with an increase in
5bCLD content, or at least with a more regular occurrence of positive values. Lastly, despite the impossibility of
generalizing behaviour with the limited sampling sites and available period series, the groundwater data sets
showed an interesting evolution pattern with, in some cases, a decrease in CLD content associated with an
increase in water 5bCLD content.

#### 3.3.2 In SW: the pesticide concentration and ratio globally decreased

From all the available data, we observed a highly significant downward trend in mean river concentrations for
the CLD content, 5bCLD content and the 5bCLD / CLD ratio in water (a slope of -0.008, -0.028, -0.018,
respectively). It is interesting to note that the decreasing trend for the 5bCLD content was about threefold higher
than for the water CLD content.

More specifically, Figure 6 shows the evolution of water CLD content for the 14 rivers with the highest
measurement frequency. Globally, the mean Sen trend was -0.008 for the log, meaning that the CLD content was
halved after 7.5 years. Although most of the rivers showed a significant decrease in water CLD content, some of
them were characterized by a constant level of contamination (Saint Pierre, Pont RN Rouge) and even one by a
slight increase (Camping Matouba). Independently, we noted a high variation in the level of contamination.

A further analysis of temporal evolution (Figure 7) highlighted a relationship between Sen trends for CLD and
the mean water 5bCLD contents (regression p-value =0.06): the lower the water 5bCLD content, the greater the
decrease in water CLD content. A similar trend was observed for the 5bCLD / CLD ratio (regression p-
value=0.05), while the relationship was not significant for mean water CLD content. This means that the




decrease intensity did not depend on water CLD content. Figure 7a and 7b (left: sen CLD *vs.* mean CLD) shows
favourable situations at the bottom left, where strong decreases in water CLD content were associated with a low
water CLD content in SW, which gives hope for pollution mitigation. Adversely, in the situations at the top right
of the figure, the pollution level is likely to last for a long time.
Additionally, Figure 7b shows that the weakest decreases in water CLD content were partly associated with
recent (0.1 My to present) geological formations and that the highest decreases were associated with older ones.
Lastly, regarding soils, Figure 7a shows that while andosols were distributed over the entire range of Sen trends,
ferralsols and vertisols characterized large decreases in water CLD content.
To sum up, high water CLD contents decreased with low water 5bCLD contents and low 5bCLD / CLD ratios
were encountered for basins situated on old geology and mostly ferralsols or vertisols. On andosols and recent
geology, the water CLD content did not vary over the study period, and the water 5bCLD content and 5bCLD /
CLD ratio were high. These conditions define a high diversity of situations with regard to the duration of
pollution.

### 3.4 Model simulation

In order to grasp the complex fate of CLD and 5bCLD, we used the simple model presented Sect. 2.3.4. It is an
iterative leaching model investigating the theoretical fate of CLD and 5bCLD in water, accounting for CLD and
5bCLD lixiviation rates ($T_{CLD}$ and $T_{5bCLD}$), as well as the rate of CLD degradation into 5bCLD ($C_{degrad}$). Table 2
gives the results of the optimization processes in order to assess $T_{5bCLD}$ and $C_{degrad}$ from realistic values of $T_{CLD}$
and the 5bCLD / CLD ratios. Thus, according to Eq. 6 (see Sect. 2.3.4), $T_{CLD}$ may vary from 0.017 for an
andosol (And model) to 0.15 for a nitisol (Nit model), considering the respective values given by Cabidoche et
al., (2009) of 20,000 and 2,000 L kg$^{-1}$ for Koc, 0.55 and 1.1 for bulk density D, 70 and 20 g kg$^{-1}$ for soil carbon
content C, and 4,000 and 2,000 mm for annual rainfall R. We targeted the 5bCLD / CLD ratios of 0.1 and 0.025
in water (cases And1, Nit1 and And2, Nit2, respectively), which corresponded to the median 5bCLD / CLD
ratios of SW for the two groups identified in Sect. 3.1. We applied a constraint on the 5bCLD / CLD ratios in
soil, considering they should lie between 0.01 and 0.017, referring to the median value encountered for andosols
and nitisols, respectively (Clostre et al., 2015).
Figure 8 shows the results of two simulations: simulation And1 corresponds to an andosol situation with high
soil retention, and simulation Nit2 to a nitisol situation with low soil retention (Table 2). Notice that, according
to Eq. (3) and (4), Figure 8 shows the leached quantities of CLD and 5bCLD, not the concentration. However,
since the two compounds are lixiviated with the same water quantities, the shape of concentration curve and
quantity curve do not differ.
The simulation results showed that the ratio increased with time over the entire period up to a plateau (see Figure
8). A decrease in the ratio was not simulated, although a global trend was noted for our observed data on the
whole. At one sampling point, such a decrease could occur with an increase in lixiviation conditions (not
shown), which may have been linked to land use changes. More likely, it could have been an artefact due the
difficulty in determining low values near the quantification threshold.



CLD decreased exponentially in the modelling approach. The current decrease we mainly observed in SW fitted with this dynamic (linear decrease in log scale, Figure 6). Interestingly, we found that the decrease rate for andosols (simulation And2 - Figure 8) was far lower than for nitisols (simulation Nit1). This matched with the andosol situation, where no significant decrease in the river was observed.

5bCLD first increased and then decreased at the same time as CLD. This may explain why we found a 5b CLD / CLD ratio increase, whereas a 5bCLD decrease was observed. Our simulations also showed that $T_{5bCLD}$ must be higher than $T_{CLD}$ otherwise the ratio increased continuously without a plateau. Optimization processes also gave a higher value for $T_{5bCLD}$ (Table 2), given that high ratios are unlikely when $T_{CLD}$ is high (0.15) since it yields a $T_{5bCLD}$ of 1 (meaning that all 5bCLD is leached).

Lastly, despite difficulties in predicting what would happen for each location, our simulations gave interesting insights for a better understanding of the global dynamics of the 5bCLD / CLD ratio and explained some of the observations in water.

## 4 Discussion

Our results showed high spatial and temporal variability for water CLD content in SW and GW contamination. By relating water CLD content to its metabolite compound, 5bCLD, we highlighted physical conditions relative to soils and geology that may explain its variability in water, but also in the dynamics of pollution trends. We summarized our conclusions in a conceptual scheme presented below.

### 4.1 CLD is degraded and contamination decreases

First of all, the CLD content in SW matched with the areas where CLD had been applied, i.e. in banana cropping areas, irrespective of geology and soils. This was consistent with a global link between the location of contaminated soil areas and the location of contaminated rivers, as shown on a watershed scale by Della Rossa (Della Rossa et al., 2017). Surprisingly, we found that, overall, the soil type had no significant effect on water CLD content in SW, although large differences in CLD content were usually encountered in soils (Clostre et al., 2015; Devault et al., 2016). This paradoxical result was consistent with previous work showing that the most contaminated soils are not the most contaminant for water, owing to their different capacity to retain the molecule (Cabidoche et al., 2009; Levillain et al., 2012; Woignier et al., 2012). In other words, two types of soils with different CLD contents may release the same quantity of CLD into water. However, our simulations showed (see Figure 8) that over a long time scale, CLD contents in a river will quickly decrease for basins draining soils such as nitisols due to their low capacity to retain CLD.

In this environment, our results were in line with CLD degradation, being visible over a decadal time period despite its strong persistence in the environment. This was hypothesized by observing the distribution of 5bCLD / CLD ratios in water (median of 0.03; 1[st] centile of 0.006) with a far higher median and first centile value than in the commercial product Curlone© (ratio of 0.0011). This was consistent with the result obtained by Devault et al. (2016), who found high 5bCLD / CLD ratios in soils and, in particular, larger amounts of 5bCLD than should have been applied using commercial formulations.



The water CLD content in SW decreased as well as the water 5bCLD content and the 5bCLD / CLD ratio. Given the mean Sen trends of about -0.008 for CLD (see Sect. 3.3.2), it takes about 40 years to yield the threshold of 0.1 µg L$^{-1}$ during baseflow periods (flood flow periods being rarely sampled) given a current concentration of 0.5 µg L$^{-1}$ on average. This trend was higher than that expected by Cabidoche et al. (2009), maybe because the authors underestimated the degradation process, which is still not greatly documented. However, it was consistent with the results obtained by Crabit (Crabit et al., 2016) based on a storage approach that assessed the duration of CLD pollution of a river of a watershed at 60 years.

**4.2 Hypothesis relative to leaching processes**

One of the main questions in this paper was what the 5bCLD / CLD ratio represents. To answer this sensitive issue, we differentiated between three dimensions. A temporal dimension because the 5bCLD / CLD ratio is supposed to increase over time with degradation progress. A spatial dimension since the 5bCLD / CLD ratio may depend on local degradation conditions. A dynamic dimension since the 5bCLD / CLD ratio may depend on the mobility properties of both molecules, CLD and 5bCLD.

The temporal dimension was firstly related to the long application period (from 1970 to 1993 for CLD), given that land-use changes led to different application phases in the 70s and 80s and that land-use changes are correlated with soil contamination levels (Desprats et al., 2004). Secondly, comparing simulation results to measurement time series, the temporal dimension could also be grasped observing GW, if we consider that the residence time within the aquifer gives a temporal window on the water infiltration conditions (Gourcy et al., 2009; Tesoriero et al., 2007). The residence time - estimated by the water apparent age - depends on hydrogeological properties, and thus to the geological context (type of lithology and its weathering level, geometry of the geological deposits, etc.). For example, we observed that high 5bCLD / CLD ratios were mainly located in the waters of northern Martinique, where rivers drain recent geological formations. In that area, unweathered formations favour rapid transfers and thus low GW residence times of several years (Arnaud et al., 2017; Gourcy et al., 2009). Thus, in that area, GW is young and probably today mainly composed of waters that percolated in the last decade with a 5bCLD / CLD ratio close to the current 5bCLD / CLD ratio in soil leaching waters. Conversely, the highest groundwater residence times in more weathered geological formations probably characterize older GW (residence time of several decades) where the 5bCLD / CLD ratio may reflect an earlier 5bCLD / CLD ratio in soil leaching waters - closer to the ratio in the commercial product - during periods of application or just several years after, leading to lower 5bCLD / CLD ratios in water.

The spatial dimension is hard to grasp since some of the variability can be attributed to the spatio-temporal variability of land-use changes over the application period. Considering that soil might be an important factor, the results from Clostre et al. (2015) show that distribution of the 5bCLD / CLD ratio weakly differs from one soil to another, with a median value of around 0.011 [0.002 0.077] in andosols and 0.017 [0.007 0.081] in nitisols. This does not mean that degradation does not depend on soil, but it does mean that we cannot assess it. It is interesting to note that the simulations accounting for nitisols and andosols in Table 2 give close values of 0.14% and 0.16% of degradation rate, respectively. The soil factor could therefore not be considered decisive in explaining spatial degradation intensity.





For the dynamic dimension, our theoretical leaching model helped to represent how contamination evolved. On the whole, the simulations accounting roughly for andosol and nitisol conditions tallied well with our observations or with results from the literature: i) a  large decrease in CLD was associated with a low 5bCLD / CLD ratio, and ii) nitisol situations are more conducive to a contamination decrease than andosol situations, considering pollution duration as noted by Cabidoche (Cabidoche et al., 2009).

Lastly, this discussion shows that the combined role of geology and soils together may explain 5bCLD / CLD ratio levels. In a comprehensive way, we derived a conceptual scheme of water contamination on a regional scale.

**4.3 A conceptual scheme of water contamination on a regional scale**

We propose a conceptual scheme in Figure 9 to explain differences in 5bCLD / CLD ratios in water. We first assumed that degradation occurs in soils. This process, which is combined with other processes determining CLD and 5bCLD fate in soil, results in a general increase in water 5bCLD content and in the 5bCLD/CLD ratio, which is more or less pronounced depending on the soil. Hydrogeology teaches us that SW today could either be a signal of ancient infiltrations and transfers underground, several decades ago, when 5bCLD/CLD ratios in soils were low (long residence time), or a signal of recent percolations, several years ago, when 5bCLD/CLD ratios in soils were high (short residence time). Thus, soil properties and residence times both contribute to explaining the current impact on water quality in SW. This explanation is consistent with high 5bCLD/CLD ratios in northern Martinique on recent geological formations, and low 5bCLD/CLD ratios elsewhere. For high 5bCLD/CLD ratios in the South on vertisols, we can speculate that the degradation process was greater in this soil type (like soil 2 in Figure 9) because lixiviation is lower in the southern area with a lower rainfall rate. This may explain the higher 5bCLD/CLD ratios in SW, as simulated by a previous model, despite a longer residence time in the aquifers.

All of these results identify a set of conditions that favour the disappearance of CLD from the environment, namely ferralsols with low retention properties on older geological formations, while others - notably andosols with high retention rates on recent formations - are more risky.

**5 Conclusion**

The aim of this paper was to identify conditions that are conducive to a decrease in organochlorine pollution levels in Martinique (FWI). We adopted an unusual approach that accounted, on the one hand, for the interactions between aquifers and rivers on a catchment scale and, on the other hand, for the fate of CLD and its compound 5bCLD. This approach was fruitful and led to the proposal of a global scheme of water contamination on a regional scale accounting for physical conditions relative to soils and geology. This scheme coherently links the various amounts of chlordecone (CLD) and its metabolite 5bCLD in SW and GW. It explains both their variability in water, but also in the dynamics of pollution trends.

Our results have several implications for evaluating diffuse pollution of agricultural origin. The spatial analysis of metabolite/parent compounds provided some interesting information for identifying risky areas, or areas where persistent pollutants are more likely decreasing. This also provided some insights into key parameters that control these conditions and environmental vulnerability to agricultural pollution. This led to implications



regarding where and how to act to reduce impacts. Another implication is to promote continuous long-term
observations as opposed to one-off sampling, completing modelling approaches: in our case, long CLD time
series revealed a faster decrease than that expected by previous model predictions. Lastly, such a spatial and
temporal overview is required on a large scale to help stakeholders manage pollution on a territory scale,
accounting for the main characteristics of the landscape. This is the main challenge for the OPA-C Observatory
in FWI (Cattan et al., 2017).
**Acknowledgments**
We are grateful to the Water Office, the general council, and the environment, planning and housing agency in
Martinique for the data they provided on surface waters. Data sets on groundwater were provided by BRGM. We
should like to thank the LDA26 and BRGM laboratories for pesticide analyses.



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





**Table 1: Effects of physical conditions on the contamination level of surface water (model 1) and groundwater (model 2), showing probability levels of tested factors**


|  | CLD | 5b | ratio |
|---|---|---|---|
| Model 1: surface water | | | |
| Soil x geology | 0.7210 | 0.5989 | **<0.0001** |
| Soil x geology x Hydrographic sector | 0.9077 | 0.1377 | **<0.0001** |
| Date | **<0.0001** | **<0.0001** | **<0.0001** |
| | | | |
| Model 2 : groundwater | | | |
| Soil *(or geology)* | **0.0228** | 0.8143 | 0.1209 |
| Soil *(or geology)* x Hydrographic sector | *0.0674* | 0.2811 | 0.6333 |
| Date | **<0.0001** | **<0.0001** | **<0.0001** |

**Bold**: statistically significant at the 0.05 probability level
*Underlined italics*: statistically significant at the 0.10 probability level




**Table 2: CLD (Cdegrad) degradation rate and 5bCLD (T5bCLD) lixiviation rate stemming from optimization**
**processes based on two target values of the 5bCLD / CLD ratio in leaching water (cases 1 and 2) and two hypotheses**
**for the CLD (TCLD) lixiviation rate account**

| Hypothesis | Target | Parameters | Optimization results | |
|---|---|---|---|---|
| | 5bCLD / CLD ratio | $T_{CLD}$ | $C_{degrad}$ | $T_{5bCLD}$ |
| And1 | 0.025 | 0.017 | 0.0010 | 0.0988 |
| And2 | 0.1 | 0.017 | 0.0014 | 0.1242 |
| Nit1 | 0.025 | 0.15 | 0.0016 | 0.2584 |
| Nit2 | 0.1 | 0.15 | 0.0126 | 1 |






**8 Figure captions**

**Figure 1: Location and relief of the island of Martinique (FWI) in the Caribbean**

**Figure 2: Relation between CLD and 5bCLD means at each sampling point for Surface Water and Groundwater; distributions of the mean of the 5bCLD / CLD ratio are given below the 2D plot.**

**Figure 3: Water CLD content (top) and 5bCLD / CLD ratio (bottom) distributions for SW (square) and GW (star), according to hydrological sectors (left), soils (middle) adapted from Colmet Daage (1965), and geology (right) adapted from Germa et al. (2011). Large squares are relative to sample points having more than ten sampling dates and small squares having fewer than ten sampling dates.**

**Figure 4: Mean 5bCLD / CLD ratio (natural logarithm) according to soil types and to the age of the geological formations. Ferr_And, Nit_And and Vert_Ferr account for watersheds with two main types of soil, namely Ferrasols and Andosols, Nitisols and Andosols, Vertisols and Ferralsols.**

**Figure 5: CLD (top) and 5bCLD (bottom) trends in GW for the three longest time series (y scale is in natural logarithm). Soil and geology are: andosol and 0.1 My to present for the Chalvet and Chez Lelene sites; nitisol and 16.1 My to 8.5 My for the Source Morne Figue site. Sen trends and p-values show a significant CLD decrease for Chalvet and Source Morne Figue. We found 5bCLD decreased at Chez Lelene while it increased at Source Morne Figue**

**Figure 6: CLD (natural logarithm) trends in SW according to geology and soil type. Sen trend and confidence interval; p value of the Modified Mann-Kendall test for serially correlated data using the Yue and Wang variance correction approach. CLD content significantly decreasd (p value <0.05) for 10 out of 14 rivers. Thick Lines (Pont RN1 and AEP-Vive-Capot) account for high decrease (lower than percentile 0.1 of Sen trends); thin lines (Camping Macouba and Saint Pierre) account for low decrease (higher than percentile 0.9).**

**Figure 7: Sen trends of CLD vs. mean log content of CLD, 5bCLD, and 5bCLD / CLD ratio (from left to right – natural logarithm) in SW, according to a) soil, and b) geology.**

**Figure 8: Theoretical evolution of CLD and 5bCLD lixiviation, as well as the 5bCLD / CLD ratio for the two models illustrating 1) conditions for andosols with high soil retention (Model And2 in Table 2 – continuous lines) and 2) conditions for nitisols with low soil retention (Model Nit1 in Table 2- dashed line).**

**Figure 9: CLD fate in soils and residence time combined to explain 5bCLD/CLD ratio levels in SW. For SW draining GW with a long residence time, leaching occurred during the application period with a low 5bCLD/CLD ratio whatever the soil type. For SW water draining GW with a short residence time, leaching occurs nowadays from soil with a higher 5bCLD/CLD ratio depending on soils and reflecting CLD fate in soils.**





**Figure 1: Location and relief of the island of Martinique (FWI) in the Caribbean**


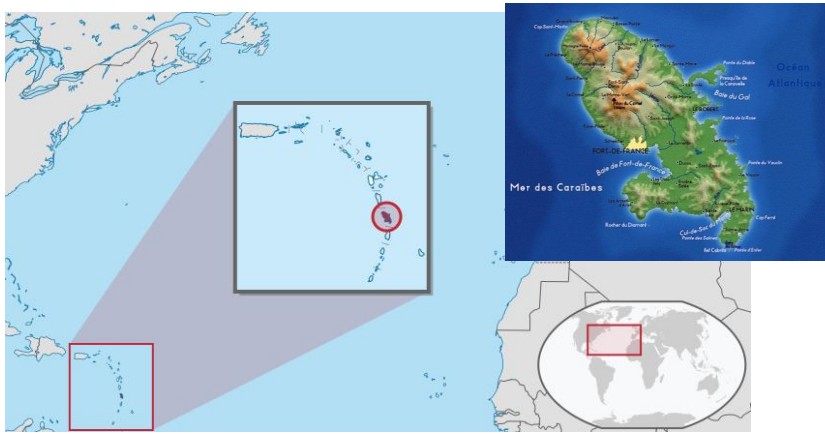








**Figure 2: Relation between CLD and 5bCLD means at each sampling point for Surface Water and Groundwater; distributions of the mean of the 5bCLD / CLD ratio are given below the 2D plot.**

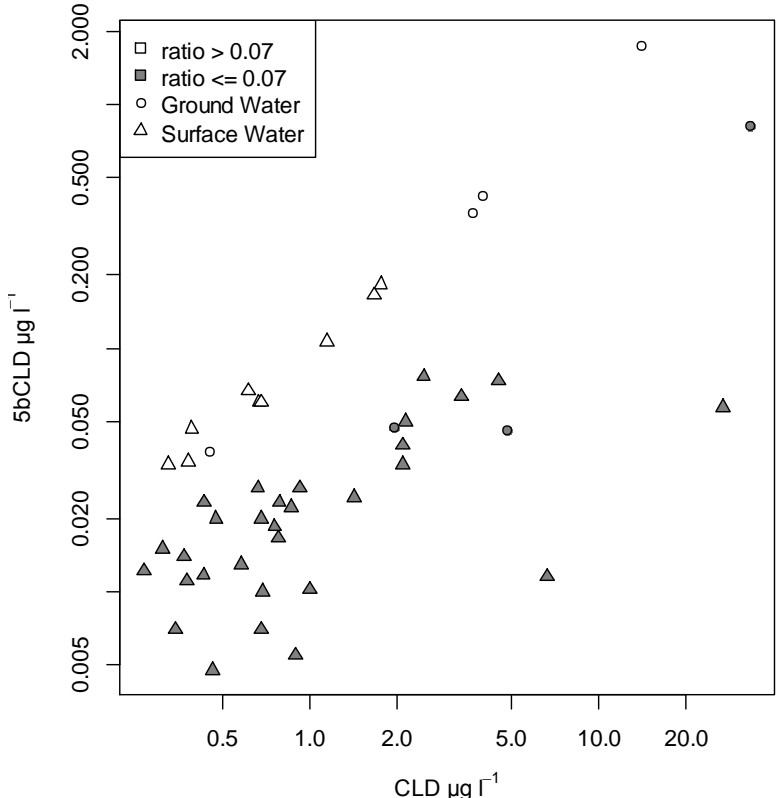

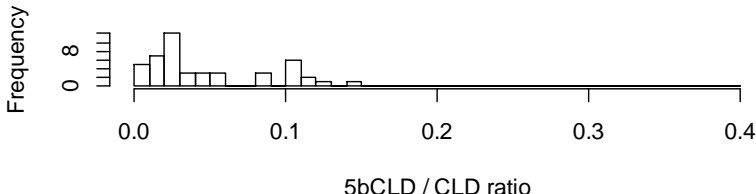







**Figure 3: Water CLD content (top) and 5bCLD / CLD ratio (bottom) distributions for SW (square) and GW (star), according to hydrological sectors (left), soils (middle) adapted from Colmet Daage (1965), and geology (right) adapted from Germa et al. (2011). Large squares are relative to sample points having more than ten sampling dates and small squares having fewer than ten sampling dates.**


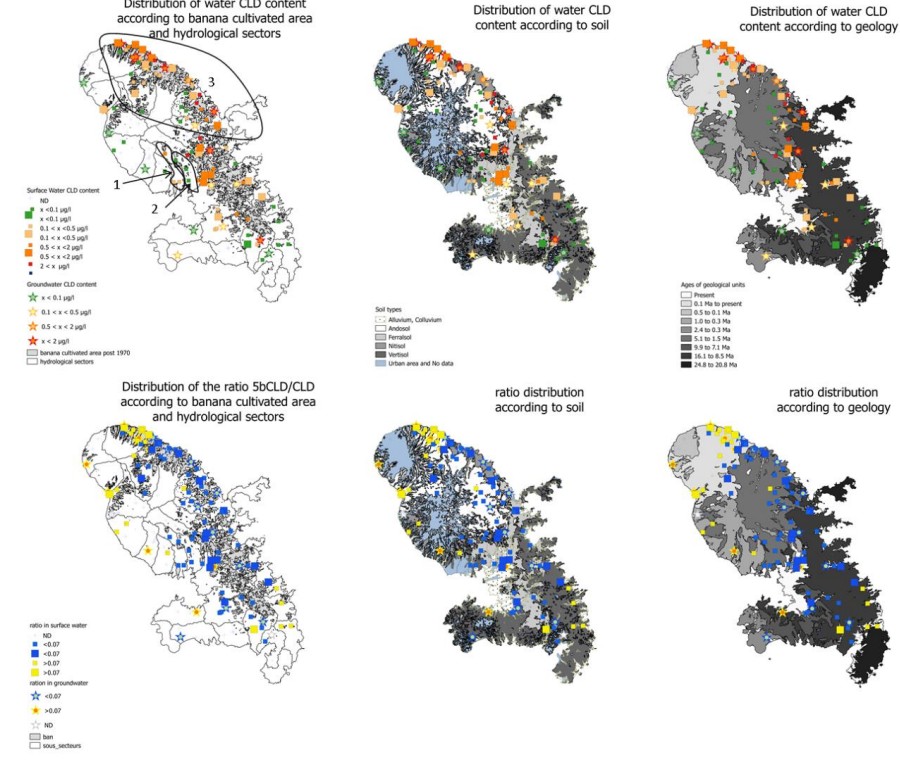




**Figure 4: Mean 5bCLD / CLD ratio (natural logarithm) according to soil types and to the age of the geological formations. Ferr_And, Nit_And and Vert_Ferr account for watersheds with two main types of soil, namely Ferrasols and Andosols, Nitisols and Andosols, Vertisols and Ferralsols.**

*Formation ages are in My*







**Figure 5: CLD (top) and 5bCLD (bottom) trends in GW for the three longest time series (y scale is in natural logarithm). Soil and geology are: andosol and 0.1 My to present for the Chalvet and Chez Lelene sites; nitisol and 16.1 My to 8.5 My for the Source Morne Figue site. Sen trends and p-values show a significant CLD decrease for Chalvet and Source Morne Figue. We found 5bCLD decreased at Chez Lelene while it increased at Source Morne Figue**

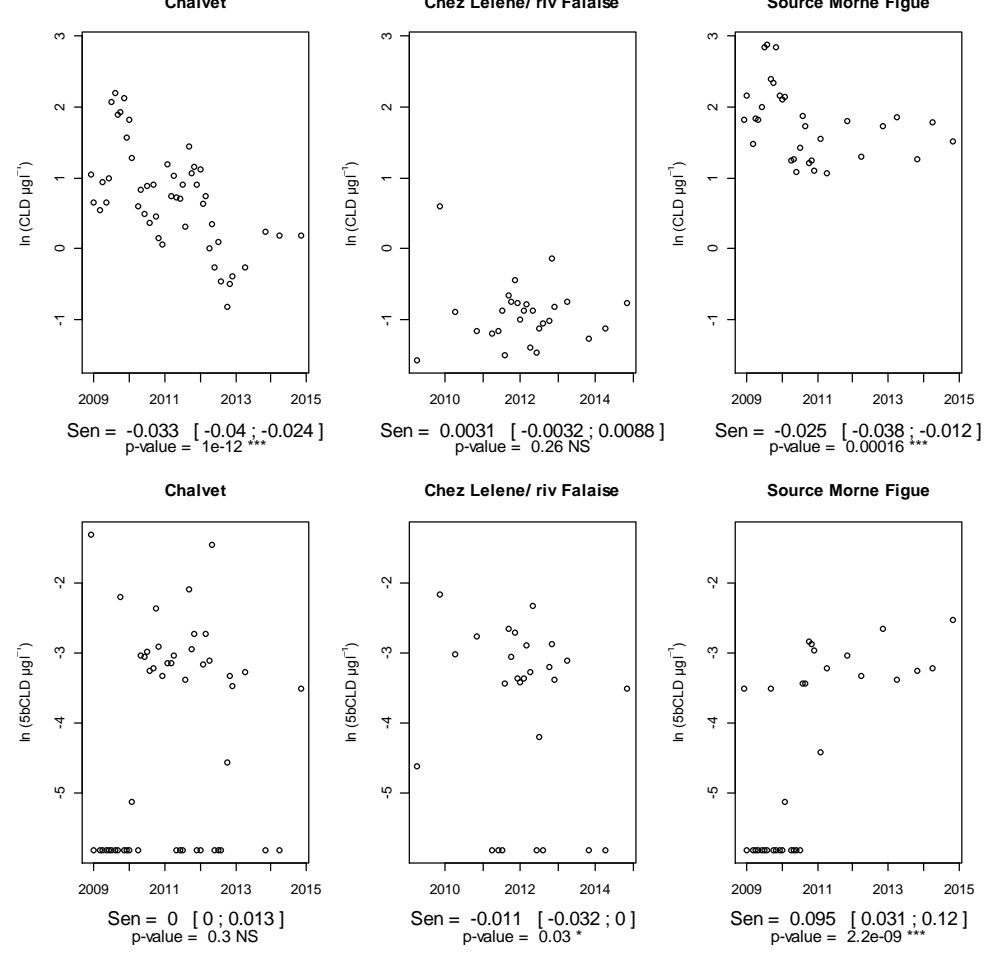







**Figure 6: CLD (natural logarithm) trends in SW according to geology and soil type. Sen trend and confidence interval; p value of the Modified Mann-Kendall test for serially correlated data using the Yue and Wang variance correction approach. CLD content significantly decreasd (p value <0.05) for 10 out of 14 rivers. Thick Lines (Pont RN1 and AEP-Vive-Capot) account for high decrease (lower than percentile 0.1 of Sen trends); thin lines (Camping Macouba and Saint Pierre) account for low decrease (higher than percentile 0.9).**


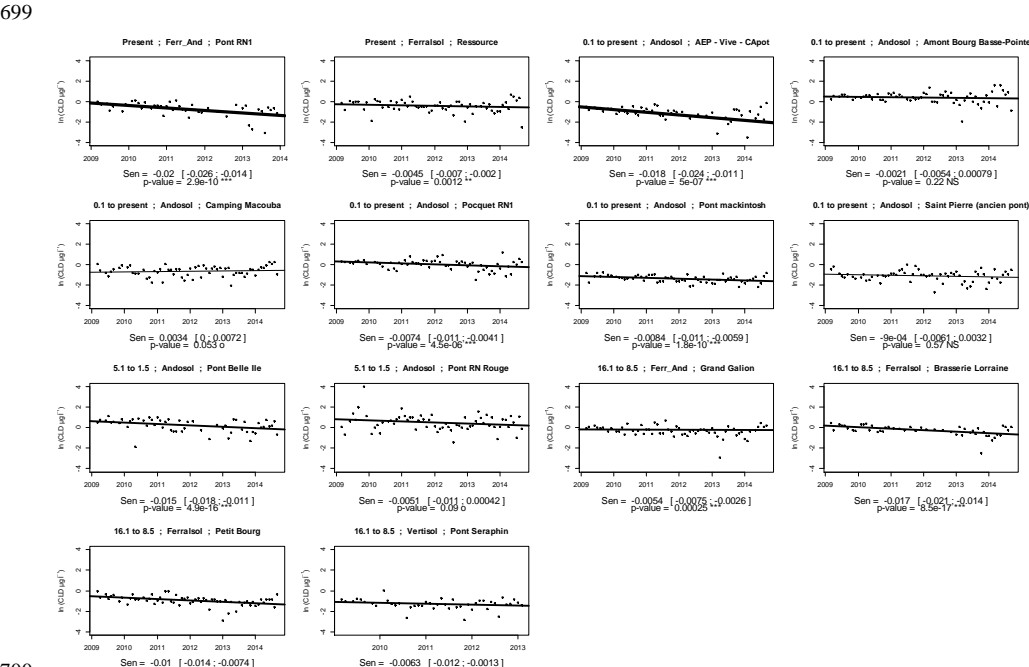






**Figure 7: Sen trends of CLD vs. mean log content of CLD, 5bCLD, and 5bCLD / CLD ratio (from left to right – natural logarithm) in SW, according to a) soil, and b) geology.**

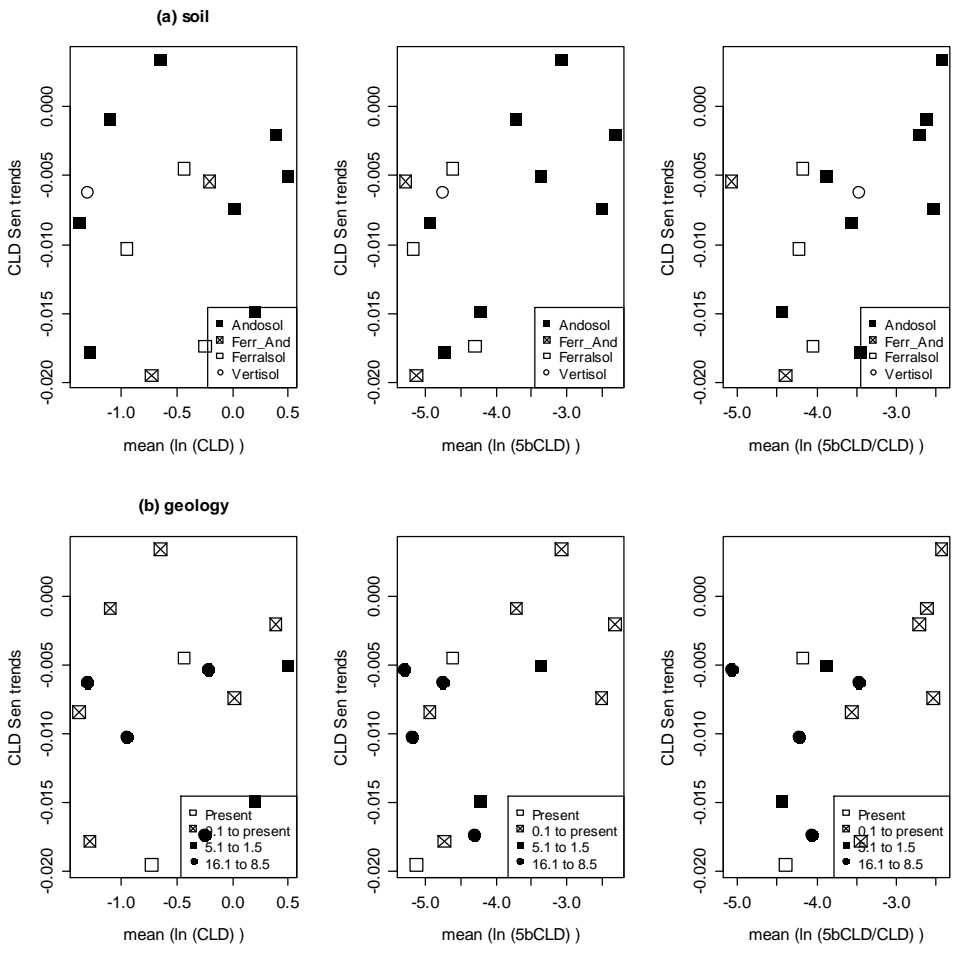








**Figure 8: Theoretical evolution of CLD and 5bCLD lixiviation, as well as the 5bCLD / CLD ratio for the two models illustrating 1) conditions for andosols with high soil retention (Model And2 in Table 2 – continuous lines) and 2) conditions for nitisols with low soil retention (Model Nit1 in Table 2- dashed line).**

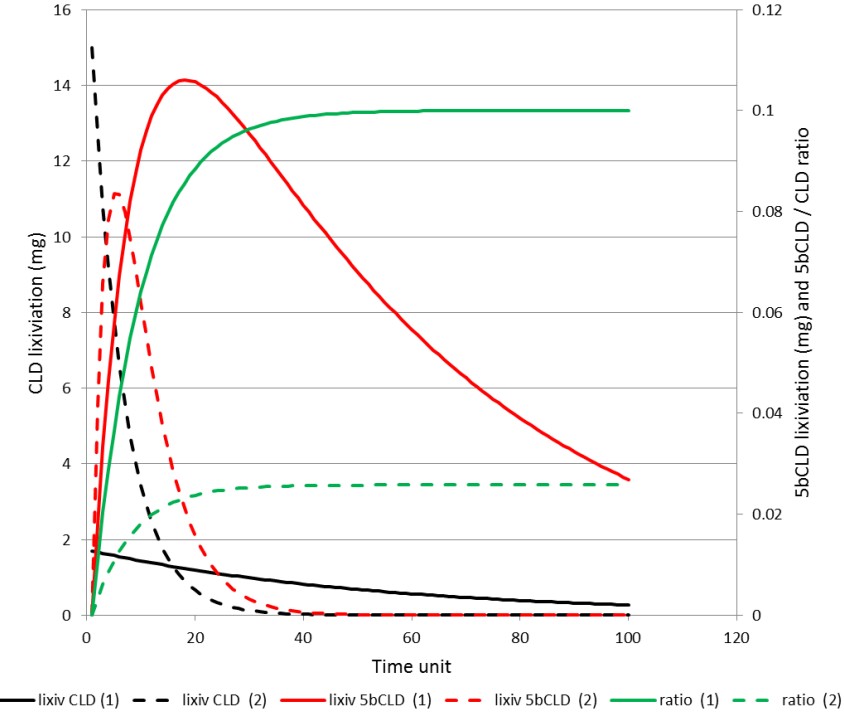





**Figure 9: CLD fate in soils and residence time combined to explain 5bCLD/CLD ratio levels in SW. For SW draining GW with a long residence time, leaching occurred during the application period with a low 5bCLD/CLD ratio whatever the soil type. For SW water draining GW with a short residence time, leaching occurs nowadays from soil with a higher 5bCLD/CLD ratio depending on soils and reflecting CLD fate in soils.**


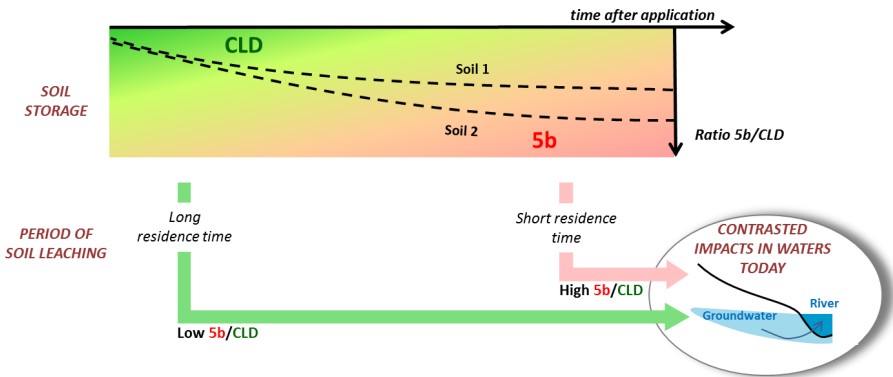


