# Peer review of "A conceptual model of organochlorine fate from a combined analysis of spatial and mid/long-term trends of surface and ground water contamination in tropical areas (FWI)"

_Hydrology and Earth System Sciences, 2018_

## Referee Comment (RC1) · S. Reichenberger (Referee) · 23 Sep 2018

Dear authors,

I think this paper is of good quality and contributes significantly to the understanding of the environmental fate of CLD on the island of Martinique. However, I have a few comments (see below) that should be addressed in order to make the paper easier to read and understand.

General comments:
1) The word "residence time" is somewhat vague. Please specify at first mention what is exactly meant (e.g. mean transit time). 2) You mention in l. 432 that the 5bCLD/CLD ratio in the commercial product Curlone was 0.0011. How was the ratio in the product Kepone that was applied before? Was the 5bCLD/CLD ratio in the commercial products constant over time, or did it vary between batches of the same product? 3) Can your findings / model be extrapolated to other CLD-contaminated areas in the Antilles, notably the island of Guadeloupe?

Specific comments:

Abstract:

4) p. 1, l. 28: "old geology": I know what you mean, but maybe "old geological substrates" or "old geological formations" would be more appropriate? 5) p. 1, l.29: "theoretical leaching model": maybe "conceptual leaching model" (as in the manuscript title)

Introduction:

6) p. 2, l. 64-65: "acute" and "environmental" are not opposites; better use "chronic" instead of "environmental" (exposure via the environment can be acute or chronic) 7) p. 2, l. 69: "partitioning coefficient (Koc) between the sorbed part on soil organic matter": not comprehensible → needs to be rephrased 8) p. 2, l. 71: "contrasting residence times": What residence times: of water or of CLD?

Materials and Methods:

9) p. 3, l. 111-112: "ferralitic soils (latosols) → ferralsols": What is the difference between the two? The WRB system contains the reference soil groups ferralsols and plinthosols, but not "ferralitic soil (latosols)". 10) p. 6, l. 213: "measurable": maybe more precisely "quantifiable", since it refers to the LOQ 11) p. 6, l. 215: "data item" (or data point): How is this defined? Unique combination of water sample and compound? 12) p. 6, l. 230: eq. 1: explain the indices (i, j, k, l, m, t) 13) p. 6, l. 234: "totally

correlated": express more precisely 14) p. 6, l. 241: "dispersion indices": How can this quantity be interpreted? 15) p. 6, l. 242: "confidence coefficient": What is this? 16) p. 6, l. 248: "Sen trends": What are Sen trends, and what do they mean statistically? (Explain in 1 or 2 sentences.) 17) p. 8, l. 258-262: What are the dimension and unit of the lixiviation rates TCLD and T5bCLD? This does not become entirely clear from eq. 6 because of the various unit conversion factors. I end up with the unit 1/year.

Results:

18) p. 8, l. 279-281: It should be mentioned here how high the ratio 5bCLD/CLD was in the commercial products that were applied, and whether it was constant over time. 19) p. 9, l. 324-325: "shorter residence times were observed for more recent formations": (Are you referring to GW only or also to SW?) This is interesting. I would rather have expected the opposite. Can you briefly explain why hydraulic residence times (mean transit times?) are shorter in the younger geological formations of Martinique than in the older ones? 20) p. 10, l. 344: "water CLD contents below the detection limit appeared less frequently": meaning not entirely clear → rephrase 21) p. 10, l. 384: unit for bulk density is missing 22) p. 13, l. 457: "unweathered formations favour rapid transfers"; Why is that? (cf. comment on l. 324-325) 23) p. 13, p. 468: "we cannot assess it": Assess what?

Conclusions:

24) p. 13, 507-508: "This led to implications regarding where and how to act to reduce impacts": Can you elaborate on this further? Is there really anything that can be done except waiting for CLD to degrade and leach from the system?

Figures:

25) Figure 3: The figure is neat, but too small for reading the legend or for identifying much on the map. → upscale 26) Figure 4: y-axis: The numbers are difficult to interpret. Try lg or non-logarithmised numbers instead of ln. 27) Figure 4: give the unit of

formation age (million years) directly in the figure (e.g. 1.0-0.3 Ma) 28) Figure 6: too small, needs upscaling (if it takes too much space after upscaling, consider shifting it to a SI section). 29) Figure 7: If one doesn't know what Sen trends are, the figure is not understandable.

Tables:

30) Table 2: Table header needs to be rephrased to improve understandability.

Best regards,

Stefan Reichenberger

---

## Referee Comment (RC2) · Anonymous Referee #2 · 15 Oct 2018

The manuscript (HESS-18-377) entitled 'A conceptual model of organochlorine fate from a combined analysis of spatial and mid/long-term trends of surface and ground water contamination in tropical areas (FWI) reports Chlordecon contamination in surface and groundwaters in the Martinique Island and proposed a conceptual approach to investigate the persistence of Chlordecon by using one reported degradation product chlordecone-5b24 hydro (5bCLD) as a tracer of on going degradation processes. This topic is important and would naturally be of interest to readers of the HESS.

However, key conceptual assumptions have to be thoroughly discussed and signifi-

cantly improved to consider publication in HESS.

First, using a transformation product (TPs) 5bCLD as a tracer of degradation extent and associated persistence is a valuable idea, but the degradation of the TP itself is never discussed. Similarly the model seems to consider as a perfect tracer 5bCLD, i.e. without degradation. This main assumption can significantly alter the assessment of persistence done and this point is never discussed. Do you have access to CLD/5bCLD ratio in soil to have an idea of the initiale signature over time to decipher soil degradation process to those associated to surface and groundwater flowpaths? Second, residence time is used to explain the spatial variability of the ratio compounds/TPs. To support the discussion, the authors should provide existing reported information/simulation of these residence times: - to discuss spatially contrasted compounds/TPs ratio delivery by soil to ground water - to address the question of degradation of the TP itself (especially for long residence time)

Third, if the sampling effort, statistical analyses and conceptual development provided a coherent approach for groundwater (slow flowpath), I have many questions on the surface water component. The representativeness of the sampling (low frequency mainly during based-flow, if I well understood the database characteristics) is not discussed taking into account percentage of Chlordecon exported during storm event associated to tropical climat. With a large Koc, the question of Chlordecon released from soil to river by erosion during runoff event is never discussed. How these pulses can contribute to spatial and temporal patterns of chlordecon in surface water? For surface water, it could be relevant to know if the CLD concentrations correspond only to the dissolved phase or if it is a "total" concentration. Information on the filtration and purification steps are not provided in the M&M section. Can contaminated sediments in river potentially be remobilized by event and alter trend assessment in surface water? In the conceptual model, the surface runoff and the surface water to groundwater seem not considered. The choice targeting mainly leaching and not the other off-site transport is never discussed. The authors mentioned "hope for pollution mitigation" based

on statistical model, but I wonder how fast flow in river can modify this assessment.

In my opinion, the paper can't be published without strengthening of these points.

Specific remarks: L324 GW, as well as in SW fed by it. And vice et versa ? L323 The age of the main geological units was used as an indicator of hydrogeology and notably residence time in the aquifers. Could you provide evidence ? Residence time assessment form others studies ? L57 and L60, 1993 or 1992, I guess banned in 1992 but used until 1993. Please explain. L121 "they are intergrades" ? L134 unweathered formations, to several decades for old weathered formations (provide range for "Old") l139 routine basis with CLD. For (double space before For) L150 5bCLD is the main CLD co- and alteration product of CLD: what do you mean by alteration product ? Transformation / degradation product ? Please clarify ? L151 "Reference standards for CLD and 5bCLD were purchased" : provide purity degree L149 Between sampling and analysis, no information is provided on the filtration (raw water/filtrated water?) , purification ?, please add L181 was not detected (i.e. 0.001 for LDA26 or 0.003 $\mu$g L$-1$ for BRGM), and an intermediate value of 0.006 $\mu$g.L$-1$ : why is different of value provided in L177 0.003 $\mu$g.L$-1$ ? Please clarify L184 double space the 5bCLD L183 Factors. Not clear for me, how heterogeneity of upstream catchment for SW or drainage area for GW were integrated in metrics ? l204. For GW, double space Fig. 1. I suggest to modify this figure to add sampling point distribution (the different zoom levels can be significantly reduced)

L248 Kendall (MK) test. We calculated Sen trends, Sen trend ? Not defined, Instead to use Sen trend in the text, I suggest to explain the information underlined by this metric (to improve understanding for the reader) The section 3.2.1. looks like a figure caption (modify and interpret directly in this section) L300 "areas since 1970, i.e. during CLD application. Surprisingly, SW and GW contamination occurred outside these banana areas" Explanation ? other dissipation Processes ? Are the contaminated areas are downstream of banana areas ?

Fig3. Legend can not be read (too small). Fig4. two small, I suggest to merge some of them or provide in SI

L308 contamination level. For example, the CLD content of hydrographic subsector 1 (see Figure 3 left for locations) was different from hydrographic subsector 2 even though the points in each zone had the same contamination level. It is very descriptive, please provide explanaton

L 320 "This statistically confirmed"/ Missing word ?

Figure 4. provide full name under the figure Ferr_And, Nit_And. . .

L375 : "duration of pollution": persistance of pollution ?

Fig 7. Legend is hiding some point: modify. I suggest to redesign the figure 7 to improve understanding of key message for the reader (add sectors/types and assessment indicator)

Fig 8. time unit ? Years. . . As discussed in the main comments, all the model predictions seem to be dependant of persistence of the 5bCLD, how the results could be altered by considering TPs degradation.

L437 0.1 $\mu$g L-1 437 during baseflow periods (flood flow periods being rarely sampled) given a current concentration of 0.5 $\mu$g L-1 438 on average. I don't understand your assumption?

L499 "catchment scale", you used watershed during all the manuscript why changed now? "The residence time - estimated by the water apparent age: not discussed or characterized before?

L388 "they should lie": sentence ?

---

## Author Comment (AC1) · 26 Oct 2018

For each comment, the response to the Referees is structured as ffollows: (1) comments from Referees, (2) author's response, (3) author's changes in manuscript in quotation marks

General comments:

1) The word "residence time" is somewhat vague. Please specify at first mention what is exactly meant (e.g. mean transit time).

[Figure]

Response: the groundwater residence time refers to the water age in aquifers defined as the mean transit time by Maloszewski and Zuber (1992). It would be addressed in the revised version of the manuscript.

Change in manuscript: L70 "Moreover, at depth, contrasting residence times of water (the water age in aquifers defined as the mean transit time (Małoszewski and Zuber, 1982)) in aquifers of several years to several decades partly account for the variability in GW contamination by CLD (Gourcy et al., 2009)"

2) You mention in l. 432 that the 5bCLD/CLD ratio in the commercial product Curlone was 0.0011. How was the ratio in the product Kepone that was applied before? Was the 5bCLD/CLD ratio in the commercial products constant over time, or did it vary between batches of the same product?

Response: according to Devault et al. (2016), Kepone and Curlone products had very similar ratio characterized by values of 0.00077. In our text, we specified a value form Clostre et al (2015). We propose to change to Devault et al. (2016) for consistency. Notice that Devault et al. also compared three different batches of Curlone without significant difference. Equally, we modified paragraph 3.1 according to your comment N°18

Change in manuscript: L430 "This was hypothesized by observing the distribution of 5bCLD / CLD ratios in water (median of 0.03; 1st centile of 0.006) with a far higher median and first centile value than in the commercial products Kepone$^{®}$ and Curlone$^{®}$ used in FWI (mean ratio of 0.00077 $\pm$ 0.00027(Devault et al., 2016)).

3) Can your findings / model be extrapolated to other CLD-contaminated areas in the Antilles, notably the island of Guadeloupe?

Response: indeed, our conceptual model could be generalized to Guadeloupe archipelago, where chlordecone was also applied intensively in banana plantations.

Change in manuscript : L441 end of §"Our results in Martinique island could indeed

be extrapolated to other CLD-contaminated areas as in Guadeloupe archipelago (FWI) where CLD was also intensively applied in banana plantations."

Specific comments: Abstract: 4) p. 1, l. 28: "old geology": I know what you mean, but maybe "old geological substrates" or "old geological formations" would be more appropriate?

Response: we agree with comment and "old geological formations" is indeed more appropriate in the abstract. In the revised version,

Change in manuscript : L28 and L373

5) p. 1, l.29: "theoretical leaching model": maybe "conceptual leaching model" (as in the manuscript title)

Responses: We agree. Change OK

Introduction: 6) p. 2, l. 64-65: "acute" and "environmental" are not opposites; better use "chronic" instead of "environmental" (exposure via the environment can be acute or chronic)

Response: We agree. Change OK

7) p. 2, l. 69: "partitioning coefficient (Koc) between the sorbed part on soil organic matter": not comprehensible ! needs to be rephrased

Response: OK. We propose to rephrase as follows, just giving the name of the coefficient: "soil organic carbon-water partitioning coefficient (Koc)"

8) p. 2, l. 71: "contrasting residence times": What residence times: of water or of CLD?

Responses: it refers to residence time of water and it will be addressed in the revised version.

Change in manuscript L70 See our response to comme 1) where we propose to add the following sentence: ". . .constrating residence times (the water age in aquifers defined

as the mean transit time (Małoszewski and Zuber, 1982)) in aquifers of. . ."

Materials and Methods: 9) p. 3, l. 111-112: "ferralitic soils (latosols) ! ferralsols": What is the difference between the two? The WRB system contains the reference soil groups ferralsols and plinthosols, but not "ferralitic soil (latosols)".

Response: It is a mistake. The climate sequence described in Comet-Daage et al, 1965 is latosols -> ferrisols -> vertisols. A correspondence with the WRB system is given by Delvaux and Brahy (2014) and is ferralsol -> nitisol -> vertisol. We corrected.

Change in manuscript : L112-112

10) p. 6, l. 213: "measurable": maybe more precisely quantifiable", since it refers to the LOQ

Response: We agree. Change OK

11) p. 6, l. 215: "data item" (or data point): How is this defined? Unique combination of water sample and compound?

Response: OK. We propose to specify in the text "data item (i.e. water samples analyzed for CLD and 5bCLD)"

12) p. 6, l. 230: eq. 1: explain the indices (i, j, k, l, m, t)

Response: There is a mistake in the text. We corrected and specified in the text:

Change in manuscript: L 233 end of §". Indices i, j, t, k, l represent factors respectively for factors soil x geology, hydrological sector, date, sampling point, sample replication"

13) p. 6, l. 234: "totally correlated": express more precisely

Response: because of a limited number of sites for groundwaters, there is not a so high spatial variability of geology and soil as observed for surface water data set. Thus, for groundwater data set (model 2), geology and soil are correlated: Andosol on recent geological formations, and ferralsols on old geological formations. To get clearer, we

modified the sentence as follows:

Change in manuscript: L234 "Soil and geological factors were closely linked for the GW data set (andosols were always associated with recent geological formations and ferralsols with old geological formations)…"

14) p. 6, l. 241: "dispersion indices": How can this quantity be interpreted? and 15) p. 6, l. 242: "confidence coefficient": What is this?

Response: dispersion indices can be interpreted like standard deviation. From the log values ln(x), we determined a confidence interval [lninf, lnsup] (with lninf = ln(x)-sd, lnsup = ln(x)+sd, sd the standard deviation of log value). That is real values are included in the interval [exp(lninf), exp(lnsup)]. Because exp(sd ) is not convenient, we defined a new dispersion index: [exp(lnsup)-exp-ln(inf)]/2, i.e. "half the difference between the limits of the confidence interval".

Change in manuscript: In fact, since we no more used these dispersion indices in the current version of the article, we propose to suppress L241 and 242.

16) p. 6, l. 248: "Sen trends": What are Sen trends, and what do they mean statistically? (Explain in 1 or 2 sentences.) Response: OK. We propose the following change

Change in manuscript: L248 "We calculated Sen trends (Sen's slope estimator, (Gilbert, 1987)) for each variable (CLD, 5bCLD and ratio) in order to compare dynamics for the two compounds. The Sen trends of a set of two-dimensional points (xi,yi) is the median m of the slopes $(yj − yi)/(xj − xi)$ determined by all pairs of sample points. The Sen's slope estimator is more robust than the least-squares estimator because it is much less sensitive to outliers"

17) p. 8, l. 258-262: What are the dimension and unit of the lixiviation rates TCLD and T5bCLD? This does not become entirely clear from eq.6 because of the various unit conversion factors. I end up with the unit 1/year. Responses: Tcld and T5bcld are

mass ratio. We propose to add a comment for that in the text:

Change in manuscript: L258 "TCLD and T5bCLD are the rates of lixiviation for CLD and 5bCLD (i.e. the ratio of lixiviated mass of CLD or 5bCLD to their respective mass in soil). . ."

Additionally we propose to modify Eq. (6) for consistency with regard to the dimensions Eq. (6): $T\_CLD = (R \times S)/(Koc \times (C/1000) \times D \times (d \times S))$ where Koc (L kg-1) is the partitioning coefficient between the sorbed part on soil organic matter and the dissolved part in water, D (kg dm-3) the bulk density, C (g kg-1) the soil carbon content, and R (dm) the annual amount of rainfall, S the soil surface (dm$^2$) et d the soil depth (dm).

Results: 18) p. 8, l. 279-281: It should be mentioned here how high the ratio 5bCLD/CLD was in the commercial products that were applied, and whether it was constant over time.

Response: We propose to add the following text at the end of the paragraph:

Change in manuscript: L281 end of §"According to Devault et al. (2016), these differences cannot stem from the use of different commercial products or different batches of a same product. Indeed, these authors found no significant statistical difference between the ratio of the commercial products Kepone$^®$ and Curlone$^®$ used in FWI, no more than between samples from different batches of Curlone$^®$. They found a mean ratio in commercial products of $0.00077 \pm 0.00027$, i.e. ten times lower than our observation in river."

19) p. 9, l. 324-325: "shorter residence times were observed for more recent formations": (Are you referring to GW only or also to SW?) This is interesting. I would rather have expected the opposite. Can you briefly explain why hydraulic residence times (mean transit times?) are shorter in the younger geological formations of Martinique than in the older ones?

Response: We refer to groundwaters (knowing that groundwaters are the main contributor of contamination of surface waters). We propose to add the following text in the 2.1 Section:

Change in manuscript: L134 after "old weathered formations". "Knowing that weathering of geological formations increases with their age, it is the main cause of a global decrease of the aquifer permeability, notably in volcanic regions (Lachassagne et al., 2014). Indeed, clayey alteration products by weathering constrain soils physical and hydrodynamic properties by reducing the porosity and consequently the permeability (Adelinet et al., 2008)."

Response: Thus, as we observe higher 5b/CLD ratio on younger geological formations (i.e. unweathered formations), we hypothesis that this was related to a shorter residence time. We propose to modify the sentence as follows: L324 Change in manuscript: "Thereby, shorter residence times were observed for aquifers located in more recent and unweathered geological formations"

20) p. 10, l. 344: "water CLD contents below the detection limit appeared less frequently": meaning not entirely clear ! rephrase

Response: We propose the following text:

Change in manuscript: L343 "For the two sites showing a decrease in water CLD content, the number of samples with 5bCLD contents below the detection limit decrease over time, and equaled zero in the case of the Source Morne Figue site after 2011"

21) p. 10, l. 384: unit for bulk density is missing

Response: We corrected (kg dm-3)

22) p. 13, l. 457: "unweathered formations favour rapid transfers"; Why is that? (cf. comment on l. 324-325)

Response: please see our response to comment "19)"

23) p. 13, p. 468: "we cannot assess it": Assess what?

Response: the effect of soil on degradation process. We modified the text accordingly.

Conclusions: 24) p. 13, 507-508: "This led to implications regarding where and how to act to reduce impacts": Can you elaborate on this further? Is there really anything that can be done except waiting for CLD to degrade and leach from the system?

Response: We propose some examples:

Change in manuscript: L508 "(e.g. choice of crops according to pollution levels since some plants are less sensitive to contamination than others (Clostre et al., 2015), constraints on water use like irrigation, choice of priority areas to test decontamination processes, setting up compensation plans according to the risk...)"

Figures: 25) Figure 3: The figure is neat, but too small for reading the legend or for identifying much on the map. ! upscale

Response: We propose a new Figure

Change in manuscript: change of Figure 3 and relative caption: "Distribution of water CLD content (a, c, e) and 5bCLD / CLD ratio (b, d, f) for SW (square) and GW (star), according to banana cultivated areas and hydrological sectors (a and b), soils (c and d) adapted from Colmet Daage (1965), and geology (e and f) adapted from Germa et al. (2011). Large squares are relative to sample points having more than ten sampling dates and small squares having fewer than ten sampling dates

26) Figure 4: y-axis: The numbers are difficult to interpret. Try lg or non-logarithmised numbers instead of ln.

Response: As specified in material and methods, data were log transformed for all analysis. For Figure 4 we worked with log-transformed data. We propose to complete the caption Figure 4 specifying the correspondence between log and non-log values:

Change in manuscript: "The y values of -6, -4 and -2 correspond to ratio values of 0.002, 0.018 and 0.135 respectively."

27) Figure 4: give the unit of formation age (million years) directly in the figure (e.g. 1.0-0.3 Ma)

Response: This is done

28) Figure 6: too small, needs upscaling (if it takes too much space after upscaling, consider shifting it to a SI section).

Response: We propose a new Figure

29) Figure 7: If one doesn't know what Sen trends are, the figure is not understandable.

Response: This is now explain in the text accordingly to the response to your comment N°16

Tables: 30) Table 2: Table header needs to be rephrased to improve understandability.

Response: We propose the following headers that matches terms in the caption and the text:

Change in manuscript: Simulation , target value, fixed parameter

Best regards, Stefan Reichenberger

References:

Adelinet M., J. Fortin, N. d'Ozouville, S. Violette, 2008. The relationship between hy-drodynamic properties and weathering of soils derived from volcanic rocks, Galapagos Islands (Ecuador). Environ Geol (2008) 56:45–58, DOI 10.1007/s00254-007-1138-3.

Clostre, Florence, Philippe Cattan, Jean-Marie Gaude, Céline Carles, Philippe Letourmy, and Magalie Lesueur-Jannoyer. 2015. "Comparative Fate of an Organochlorine, Chlordecone, and a Related Compound, Chlordecone-5b-Hydro, in Soils and Plants." Science of The Total Environment 532 (November): 292–300. https://doi.org/10.1016/j.scitotenv.2015.06.026.

Delvaux, B.; Brahy, V. "Mineral Soils conditioned by a Wet (Sub)Tropical Climate". FAO.

Retrieved 14 June 2014.

Devault, D. A., Laplanche, C., Pascaline, H., Bristeau, S., Mouvet, C. and Macarie, H. 2016. Natural transformation of chlordecone into 5b-hydrochlordecone in French West Indies soils: statistical evidence for investigating long term persistence of organic pollutants, Environ. Sci. Pollut. Res., 23(1), 81–97, doi:10.1007/s11356-015-4865-0.

Gilbert, Richard O. 1987. Statistical Methods for Environmental Pollution Monitoring. New York: Wiley.

Lachassagne P., B. Aunay,N. Frissant, M. Guilbert, and A. Malard, 2014. High-resolution conceptual hydrogeological model of complex basaltic volcanic islands: a Mayotte, Comoros, case study. Terra Nova, 26, 307–321, doi: 10.1111/ter.12102.

Maloszewski, P., Zuber, A., 1982. Determining the turnover time of groundwater systems with the aid of environmental tracers: I. : Models and their applicability, J. Hydrol., 57, 207-231.
* * *
[Figure]

**Fig. 1.**

[Figure]

**Fig. 2.**

**Present ; Ferr_And ; Pont RN1**

Sen = −0.02  [ −0.026 ; −0.014 ]
p−value = 2.9e−10 ***

**Present ; Ferralsol ; Ressource**

Sen = −0.0045  [ −0.007 ; −0.002 ]
p−value = 0.0012 **

**0.1 to present ; Andosol ; AEP − Vive − CApot**

Sen = −0.018  [ −0.024 ; −0.011 ]
p−value = 5e−07 ***

**0.1 to present ; Andosol ; Amont Bourg Basse−Poi**

Sen = −0.0021  [ −0.0054 ; 0.00079 ]
p−value = 0.22 NS

**0.1 to present ; Andosol ; Camping Macouba**

Sen = 0.0034  [ 0 ; 0.0072 ]
p−value = 0.053 o

**0.1 to present ; Andosol ; Pocquet RN1**

Sen = −0.0074  [ −0.011 ; −0.0041 ]
p−value = 4.5e−06 ***

**0.1 to present ; Andosol ; Pont mackintosh**

Sen = −0.0084  [ −0.011 ; −0.0059 ]
p−value = 1.8e−10 ***

**0.1 to present ; Andosol ; Saint Pierre (ancien por**

Sen = −9e−04  [ −0.0061 ; 0.0032 ]
p−value = 0.57 NS

**5.1 to 1.5 ; Andosol ; Pont Belle Ile**

Sen = −0.015  [ −0.018 ; −0.011 ]
p−value = 4.9e−16 ***

**5.1 to 1.5 ; Andosol ; Pont RN Rouge**

Sen = −0.0051  [ −0.011 ; 0.00042 ]
p−value = 0.09 o

**16.1 to 8.5 ; Ferr_And ; Grand Galion**

Sen = −0.0054  [ −0.0075 ; −0.0026 ]
p−value = 0.00025 ***

**16.1 to 8.5 ; Ferralsol ; Brasserie Lorraine**

Sen = −0.017  [ −0.021 ; −0.014 ]
p−value = 8.5e−17 ***

**16.1 to 8.5 ; Ferralsol ; Petit Bourg**

Sen = −0.01  [ −0.014 ; −0.0074 ]
p−value = 5.6e−11 ***

**16.1 to 8.5 ; Vertisol ; Pont Seraphin**

Sen = −0.0063  [ −0.012 ; −0.0013 ]
p−value = 0.0071 **

**Fig. 3.**

**(a) soil**

**(b) geology**

**Fig. 4.**

---

## Author Comment (AC2) · 26 Oct 2018

The response to the Referees is structured as follows: (1) comments from Referees, (2) author's response, (3) author's changes in manuscript in quotation marks

RC: First, using a transformation product (TPs) 5bCLD as a tracer of degradation extent and associated persistence is a valuable idea, but the degradation of the TP itself is never discussed. Similarly the model seems to consider as a perfect tracer 5bCLD, i.e. without degradation. This main assumption can significantly alter the assessment of

persistence done and this point is never discussed.

Response: We thank the reviewer for raising this important point. Degradation of the TP can be discussed adding a new calculation step in the model accounting for 5bCLD degradation. Eq (4) can be modified as follow: 5bCLD(t+1)=5bCLD(t)-5bCLD(t)×T_5bCLD-5bCLD(t)×C_5bdegrad+CLD(t)×C_degrad

Then, we test 3 values of C5bdegrad in a wide range surrounding the one of CLD (Cdegrad): C5bdegrad = 0, C5bdegrad= Cdegrad, C5bdegrad =10 x Cdegrad. Notice these values are highly speculative since there is no experimental value of C5bdegrad and that Cdegrad is the result of optimization process in our paper. Results are reported in the Figure "test" which shows the evolution of the 5bCLD lixiviation and of the ratio for the 3 tested values. The figure 1 shows similar dynamics of ratio evolution or of lixiviation evolution. The difference between the simulations remains weak, notably because the tested values are about 10 and 100 times lower than lixiviation rate (T5bCLD equals 0.1242 here while Cdegrad equals 0.0014)). Consequently, introducing a degradation coefficient does not alter here our first conclusions. Running optimization process with this new term we find T5bCLD = 0.1242 ; Cdegrad = 0.0014 ; C5bdegrad = 0.0010 Our assumptions are consistent with estimations of Dolfing et al. (2012) showing that the solubility is higher for transformation products of CLD.

Change in manuscript: So, to account for the reviewer comment, we propose 1) To complete the current model adding a degradation term for 5bCLD. 2) Given the lack of knowledge and the uncertainty about degradation rate, we propose to add the following comment in the text. "The values of degradation remain uncertain since we have no reference for comparison. In our case, the optimization process yield a far lower degradation rate compare to the lixiviation rate. Consequently, the model will be less sensitive to changes in degradation rate than in lixiviation rate that determine the ratio in water. Additionally, there is an uncertainty comparing degradation rates for 5bCLD and CLD. Optimization process yield degradation rates for 5bCLD and CLD of the same order of magnitude. Additional simulations show that setting C5bdegrad ten

times higher than Cdegrad instead of zero reduce the ratio 5bCLD / CLD by 10 percent without changing the dynamic of ratio and of 5bCLD lixiviation (not shown). Knowing that transformation products of chlordecone are likely to be more mobile in the environment than their parent compound (Dolfing et al. 2012), we assume our model give sufficient bases for interpreting our results".

RC: Do you have access to CLD/5bCLD ratio in soil to have an idea of the initial signature over time to decipher soil degradation process to those associated to surface and groundwater flowpaths?

Response: Reference of ratio in soils are in the paper of Clostre et al (2015).The median value of 0.011 in nitisols and 0.017 in andosols were used in our paper to constrain our model (see section 3.4). This does not help to speculate about ratios in water since they depend on lixiviation rates of CLD and 5bCLD. In our article, data from Cabidoche et al (2009) were used to assess CLD lixiviation rate (TCLD) for andosols and nitisols. The 5bCLD lixiviation rate (T5bCLD) stemming from the optimization process appears higher than TCLD. This result is consistent with Devault et al (2016) who conclude for a higher mobility for 5bCLD than for CLD. Whatever, it is unlikely that CLD was leached while 5bCLD accumulated in soil profile due to the highest mobility of transformation products (Dolfing et al., 2012).

Change in manuscript: We propose to add the following sentence section 3.4 L406: "... continuously without a plateau. This result is consistent with Devault et al (2016) who conclude for a higher mobility for 5bCLD than for CLD, and more generally with results of Dolfing et al. (2012) who shows that transformation products have a highest mobility than CLD."

RC: Second, residence time is used to explain the spatial variability of the ratio compounds/TPs. To support the discussion, the authors should provide existing reported information/simulation of these residence times: - to discuss spatially contrasted compounds/TPs ratio delivery by soil to ground water

Response: please see our response to the comment "19)" of the First referee

RC- to address the question of degradation of the TP itself (especially for long residence time)

Response: longer residence time does not mean that the TP degradation is higher. In fact degradation occurs in the soil, whereas residence time in the aquifer refers to transfers in depth (below soil cover) where the degradation (as well as the retention) is considered as null. Groundwater residence time is generally superior to several years (up to several decades – see Gourcy et al., 2009 for instance) that is widely superior to the residence time of the infiltrated water in the soil cover (several days or months).

RC: Third, if the sampling effort, statistical analyses and conceptual development provided a coherent approach for groundwater (slow flowpath), I have many questions on the surface water component.

Response: Global comment about flowpath. First, volcanic soils in Caribbean islands have a high infiltration capacity (saturated hydraulic conductivity superior to 60 mm/h (Cattan et al., 2006; Crabit et al, 2016). Then, despite high rainfall intensities and amounts, most of rainfall infiltrates (about 95% at the plot scale according to Cabidoche et al, (2009); more than 90% at the watershed scale according to Charlier et al., 2008; 2011) generating either subsurface or deep flows. So leaching is the main process in pesticide transport.

Second, usually, one reason to study separately pesticide transport by surface runoff is that the pesticide concentration in runoff water may vary highly according to time of pesticide application at the plot scale (Saison et al., 2008) as well as at the watershed scale (Charlier et al., 2009). It is not the case for CLD which have been applied long time ago: boundary conditions relative to pesticide concentration in soil are almost steady. Surely, during application period, agricultural practices may have affect 5bCLD/CLD ratio day by day. However our model aims to simulate the ratio evolution over a long time period. A second reason to consider separately runoff and infiltration

water is that pesticide concentration in surface water at the plot scale may differ from infiltrated water. There are few references about this point for CLD. Cabidoche et al (2009) notice that CLD concentration in surface runoff was more than 3-fold lower than in drainage, while runoff volume was 10 times lower than drainage volume. They consequently neglected loads in runoff that represented less than 1/30 of those in drainage at the plot scale.

Given the previous consideration, we then chose to focus here on lixiviation process, which affect the ratio dynamic on the long term. The reviewer ask the question of the effect of event-driven process (storm event, surface runoff, erosion, application practices) on long term trends and how they can modify CLD concentration in water and the ratio. It is a difficult issue that would require getting spatial distribution of stormy event, and their contribution to river pollution. This lack of knowledge probably leads to minor CLD exportation. Indeed, most of the time (even in rainy regions), surface flow in the river is driven by baseflow from aquifer's drainage, originated from water infiltration. Knowing that groundwater concentrations are widely higher than in rivers, concentrations during storm events would led to generate diluted concentrations in surface waters.

We propose different changes relatively to the reviewer comments. We equally propose to add a §"main assumption about CLD transfer" in discussion section

Change in the manuscript: L417 addition of the §"4.1 main assumption about CLD transfer. In our study we focused on long term trend of CLD and 5bCLD concentration in water and their ratio. We considered that the main process that determined pollutant concentrations in water was relative to the CLD desorption by water that infiltrates into the soil. We assumed this hypothesis for different reasons. First, water mainly infiltrates. In fact given the high soil infiltration rate (saturated hydraulic conductivity superior to 60 mm/h (Cattan et al. 2006; Crabit et al, 2016), most of rainfall infiltrates (about 95% at the plot scale according to Cabidoche et al, (2009); more than 90% at the watershed scale according to Charlier et al., 2008;2011) generating

either subsurface or deep flows. Consequently, transportation by surface runoff is low. Cabidoche et al (2009) notice that CLD concentration in surface runoff was more than 3-fold lower than in drainage, while runoff volume was 10 times lower than drainage volume. They consequently neglected loads in surface runoff that represented less than 1/30 of those in drainage at the plot scale Second soils have little erodibility: Cabidoche et al (2009) notice that "All the soil types in FWI are acidic, which prevents clay dispersion and sheet erosion. Hydric erosion appears to be due only to bad soil management practices, which concentrate runoff that then forms streams that are able to carry aggregates". So erosion from cultivated soils is probably not a major way of CLD transportation. Moreover, given the torrential type flow in rivers in FWI, the most likely future of eroded soil, is to sediment in the sea, with then a weak impact on river pollution. Finally, neglecting transport via surface runoff on plots and hillslopes, we probably underestimated pollutant exportation. But we expected that it should not have a great impact on the long-term dynamics of concentrations and ratio in rivers, which is one of the main topic of our paper".

RC: The representativeness of the sampling (low frequency mainly during based-flow, if I well understood the database characteristics) is not discussed taking into account percentage of Chlordecon exported during storm event associated to tropical climat.

Response: We propose to add the following sentence:

Change in manuscript: L438 "However, since sampling mainly occurred outside storm event periods, calculation with these data will lead to minor the estimate of CLD exportations."

RC: With a large Koc, the question of Chlordecon released from soil to river by erosion during runoff event is never discussed. How these pulses can contribute to spatial and temporal patterns of chlordecon in surface water?

Response: Some studies are underway on the subject. At the moment, Cabidoche et al (2009) notice that "All the soil types in FWI are acidic, which prevents clay dispersion

and sheet erosion. Hydric erosion appears to be due only to bad soil management practices, which concentrate runoff that then forms streams that are able to carry aggregates". So erosion from cultivated soils is probably not a major way of CLD transportation. Second, it is difficult to speculate on the future of eroded contaminated soil and their impact on water contamination. Given the torrential type flow in rivers in FWI, the most likely future of eroded soil, is to sediment in the sea, with then weak impact on river pollution.

Change in manuscript: sea the new §4.1 above

RC: For surface water, it could be relevant to know if the CLD concentrations correspond only to the dissolved phase or if it is a "total" concentration.

Response: the CLD concentration is a total concentration

RC: Information on the filtration and purification steps are not provided in the M&M section.

Response: there was no purification nor filtration since the suspended matter content of samples was low (less than 250 mg L-1). Analyses were performed on raw water. We propose to add the following sentences section 2.2.2

Change in manuscript: L153 "Analyses were carried out on raw sampling water. Thus, the CLD and 5bCLD water contents correspond to the dissolved and particulate fractions. Note that the particulate fraction of the samples was low (< 250 mg L-1) due to sampling conducted mainly during periods of low flow."

RC: Can contaminated sediments in river potentially be remobilized by event and alter trend assessment in surface water?

Response: see our previous response

RC: In the conceptual model, the surface runoff and the surface water to groundwater seem not considered. The choice targeting mainly leaching and not the other off-site

transport is never discussed. The authors mentioned "hope for pollution mitigation" based on statistical model, but I wonder how fast flow in river can modify this assessment.

Response: see our previous response

Specific remarks: RC: L324 GW, as well as in SW fed by it. And vice et versa ?

Response: whereas infiltration from ditches towards aquifers is a likely process in such regions due to the high permeability of the shallow formations (Charlier, 2007), and even if some river infiltrations may contribute also to groundwater recharge (Charlier et al., 2011), we consider that the infiltration of surface water is a minor process of groundwater contamination at a global scale. In fact, in cultivated areas, surface water is generally widely less contaminated in CLD than groundwaters.

RC: L323 The age of the main geological units was used as an indicator of hydrogeology and notably residence time in the aquifers. Could you provide evidence ? Residence time assessment form others studies ?

Response: please see our response to the comment "19)" of the first referee

RC: L57 and L60, 1993 or 1992, I guess banned in 1992 but used until 1993. Please explain.

Response: Yes there was exemption until 1993. We propose

Changes in the manuscript: L60 "...ban in 1992 (there was exemption in FWI until 1993)"

RC: L121 "they are intergrades" ?

Response: Intergrades are defined by Colmet-Daage relative to the climatic sequence ferralsols -> vertisols for soils that are "intermediate". Since Colmet-Daage classification is specific, we propose to suppress the last part of the sentence which is unclear "and they are intergrades resulting from the alteration of ferralitic soils)"

RC: L134 unweathered formations, to several decades for old weathered formations (provide range for "Old")

Change in manuscript: "between a few years for recent unweathered formations (<0.5-1My), to several decades for old weathered formations (> 1My)"

RC: L139 routine basis with CLD. For (double space before For)

Response: OK

RC: L150 5bCLD is the main CLD co- and alteration product of CLD: what do you mean by alteration product ? Transformation / degradation product ? Please clarify ?

Response: In fact, 5bCLD can be considered both as a co-product and as a degradation product. Consulting biochemists, the word "alteration" seemed more convenient. We propose the following change:

Change in manuscript: "5bCLD is the main alteration product of CLD (the term "alteration" here means that 5b is both a co-product and a degradation product)for which . . ."

RC: L151 "Reference standards for CLD and 5bCLD were purchased" : provide purity degree

Change in manuscript: "with a purity degree of 96.7%."

RC: L149 Between sampling and analysis, no information is provided on the filtration (raw water/filtrated water?) , purification ?, please add

Response: OK analyses were performed on raw water

RC: L181 was not detected (i.e. 0.001 for LDA26 or 0.003 $\mu$g L$-1$ for BRGM), and an intermediate value of 0.006 $\mu$g.L$-1$ : why is different of value provided in L177 0.003 $\mu$g.L$-1$ ? Please clarify

Response: 0.003 line 177 refers to the limit of detection; 0.006 is an intermediate

value between the limit of detection 0.003 and the limit of quantification 0.01 when the compound was detected but not measurable. We propose to change "measurable" by "quantifiable"

RC: L184 double space the 5bCLD

Response: OK

RC: L183 Factors. Not clear for me, how heterogeneity of upstream catchment for SW or drainage area for GW were integrated in metrics ?

Response: factors refer to global descriptors that don't integrate such spatial heterogeneity at a local scale. Apart soil (as it is explained in the text), each site is associated with the factor value at the sampling point.

RC L204. For GW, double space

Response: OK

RC Fig. 1. I suggest to modify this figure to add sampling point distribution (the different zoom levels can be significantly reduced)

Response: sampling distribution are presented in figure 3

RC: L248 Kendall (MK) test. We calculated Sen trends, Sen trend ? Not defined, Instead to use Sen trend in the text, I suggest to explain the information underlined by this metric (to improve understanding for the reader)

Response: OK, please see our response to the first Referee. We propose the following change

Change in manuscript: L248 "We calculated Sen trends(namely Sen's slope estimator, (Gilbert, 1987)) for each variable (CLD, 5bCLD and ratio) in order to compare dynamics for the two compounds. The Sen trends of a set of two-dimensional points $(x_i, y_i)$ is the median m of the slopes $(y_j - y_i)/(x_j - x_i)$ determined by all pairs of sample points. The

Sen's slope estimator is more robust than the least-squares estimator because it is much less sensitive to outliers"

RC: The section 3.2.1. looks like a figure caption (modify and interpret directly in this section)

Response: Section 3.2.1 aims to present Figure 3 and the distribution of pollution

RC: L300 "areas since 1970, i.e. during CLD application. Surprisingly, SW and GW contamination occurred outside these banana areas" Explanation ? other dissipation Processes ? Are the contaminated areas are downstream of banana areas ?

Response: we suggest CLD misuse L304

RC: Fig3. Legend can not be read (too small).

Response: We propose a new Figure with larger legend

RC: Fig4. two small, I suggest to merge some of them or provide in SI

Response: there are two comments for Figure 4. Perhaps this comment is relative to Fig3 ? or Fig6. We propose a new Figure 6

RC: L308 contamination level. For example, the CLD content of hydrographic subsector 1 (see Figure 3 left for locations) was different from hydrographic subsector 2 even though the points in each zone had the same contamination level. It is very descriptive, please provide explanaton

Response: We propose to rephrase L308-309

Change in manuscript: "For example, although sample points of subsector 1 and 2 are very close, they do not have the same contamination level. In contrast, all sample points of subsector 1 have the same contamination level (same for subsector 2). This suggest that the hydrographic sector, i.e. the water flows within a same hydrological unit, mainly determined contamination level of sample points rather than the geographical closeness of these points."

RC: L 320 "This statistically confirmed"/ Missing word ?

Response: We propose to rephrase: "This is a statistical confirmation of the result mapped in Figure 3. . ."

RC: Figure 4. provide full name under the figure Ferr_And, Nit_And. . .

Response: OK

RC: L375 : "duration of pollution": persistance of pollution ?

Change in manuscript: "persistence of pollution"

RC: Fig 7. Legend is hidding some point: modify. I suggest to redesign the figure 7 to improve understanding of key message for the reader (add sectors/types and assessment indicator) ???

Response: We propose to keep the legend of the figures in the middle and to suppress the legend of figures on the left and right sides where points are hidden. Sen trends for others factors (hydrographic sectors and historical banana area) are not represented due to the absence of relationships.

Change in manuscript: redesign of figure 7 whith and a new caption: "Sen trends of CLD vs. mean log content of CLD, 5bCLD, and 5bCLD / CLD ratio (from left to right – natural logarithm) in SW, according to a) soil, and b) geology (for soil and geology, see legend in the middle figure).

RC Fig 8. time unit ? Years. . . As discussed in the main comments, all the model predictions seem to be dependant of persistence of the 5bCLD, how the results could be altered by considering TPs degradation.

Response: see response above

RC: L437 0.1 $\mu$g L-1 437 during baseflow periods (flood flow periods being rarely sampled) given a current concentration of 0.5 $\mu$g L-1 438 on average. I don't understand your assumption?

Response: baseflow periods refers to periods without flood flows (or storm flows). Please see also our response to your comment "34)"

RC: L499 "catchment scale", you used watershed during all the manuscript why changed now? "The residence time - estimated by the water apparent age: not discussed or characterized before?

Response: catchment is replaced by watershed. Regarding the residence time, it was discussed in L452-458 of the submitted version

RC: L388 "they should lie": sentence ?

Response: the ratios should lie

Additional references

Cattan, P., Y.-M. Cabidoche, J.-G. Lacas, and M. Voltz. 2006. Effects of tillage and mulching on runoff under banana (Musa spp.) on a tropical Andosol. Soil Tillage Res. 86:38–51.

Charlier, J.-B. 2007. Fonctionnement et modélisation hydrologique d'un petit bassin versant cultivé en milieu volcanique tropical. Ph.D. diss. Université des Sciences et Techniques du Languedoc, Montpellier II.

Dolfing J., I. Novak, A. Archelas, and H. Macarie, 2012. Gibbs Free Energy of Formation of Chlordecone and Potential Degradation Products: Implications for Remediation Strategies and Environmental Fate. Environ. Sci. Technol., 2012, 46 (15), pp 8131–8139. DOI: 10.1021/es301165p

Saison, C., P. Cattan, X. Louchart, and M. Voltz. 2008. Eff ect of spatial heterogeneities of water fl uxes and application pattern on cadusafos fate on banana cultivated andosols. J. Agric. Food Chem. 56:11947–11955.

[Figure]

[Figure]

**Fig. 1.** test

Banana cultivated area post 1970
Hydrological sectors

Soil types
Alluvium, Colluvium
Andosol
Ferralsol
Nitisol
Vertisol
Urban area and No data

Ages of geological units
Present
0.1 Ma to present
0.5 to 0.1 Ma
1.0 to 0.3 Ma
2.4 to 0.3 Ma
5.1 to 1.5 Ma
9.9 to 7.1 Ma
16.1 to 8.5 Ma
24.8 to 20.8 Ma

a)  b)

c)  d)

e)  f)

Surface Water CLD content
ND
x <0.1 µg/l
x <0.1 µg/l
0.1 < x <0.5 µg/l
0.1 < x <0.5 µg/l
0.5 < x <2 µg/l
0.5 < x <2 µg/l
2 < x  µg/l
Groundwater CLD content
x < 0.1 µg/l
0.1 < x < 0.5 µg/l
0.5 < x < 2 µg/l
x < 2 µg/l

Ratio in surface water
ND
<0.07
<0.07
>0.07
>0.07
Ratio in groundwater
<0.07
>0.07
ND

**Fig. 2.** Distribution of water CLD content (a, c, e) and 5bCLD / CLD ratio (b, d, f) for SW (square) and GW (star), according to banana cultivated areas and hydrological sectors (a and b), soils (c and d) adap

[Figure]

**Fig. 3.** Mean 5bCLD / CLD ratio (natural logarithm) according to soil types and to the age of the geological formations. Ferr_And, Nit_And and Vert_Ferr account for watersheds with two main types of soil, name

**Present ; Ferr_And ; Pont RN1**

Sen = −0.02 [ −0.026 ; −0.014 ]
p−value = 2.9e−10 ***

**Present ; Ferralsol ; Ressource**

Sen = −0.0045 [ −0.007 ; −0.002 ]
p−value = 0.0012 **

**0.1 to present ; Andosol ; AEP – Vive – CApot**

Sen = −0.018 [ −0.024 ; −0.011 ]
p−value = 5e−07 ***

**0.1 to present ; Andosol ; Amont Bourg Basse–Poi**

Sen = −0.0021 [ −0.0054 ; 0.00079 ]
p−value = 0.22 NS

**0.1 to present ; Andosol ; Camping Macouba**

Sen = 0.0034 [ 0 ; 0.0072 ]
p−value = 0.053 o

**0.1 to present ; Andosol ; Pocquet RN1**

Sen = −0.0074 [ −0.011 ; −0.0041 ]
p−value = 4.5e−06 ***

**0.1 to present ; Andosol ; Pont mackintosh**

Sen = −0.0084 [ −0.011 ; −0.0059 ]
p−value = 1.8e−10 ***

**0.1 to present ; Andosol ; Saint Pierre (ancien por**

Sen = −9e−04 [ −0.0061 ; 0.0032 ]
p−value = 0.57 NS

**5.1 to 1.5 ; Andosol ; Pont Belle Ile**

Sen = −0.015 [ −0.018 ; −0.011 ]
p−value = 4.9e−16 ***

**5.1 to 1.5 ; Andosol ; Pont RN Rouge**

Sen = −0.0051 [ −0.011 ; 0.00042 ]
p−value = 0.09 o

**16.1 to 8.5 ; Ferr_And ; Grand Galion**

Sen = −0.0054 [ −0.0075 ; −0.0026 ]
p−value = 0.00025 ***

**16.1 to 8.5 ; Ferralsol ; Brasserie Lorraine**

Sen = −0.017 [ −0.021 ; −0.014 ]
p−value = 8.5e−17 ***

**16.1 to 8.5 ; Ferralsol ; Petit Bourg**

Sen = −0.01 [ −0.014 ; −0.0074 ]
p−value = 5.6e−11 ***

**16.1 to 8.5 ; Vertisol ; Pont Seraphin**

Sen = −0.0063 [ −0.012 ; −0.0013 ]
p−value = 0.0071 **

**Fig. 4.** CLD (natural logarithm) trends in SW according to geology and soil type. Sen trend and confidence interval; p value of the Modified Mann-Kendall test for serially correlated data using the Yue and Wan

**(a) soil**

**(b) geology**

**Fig. 5.** Sen trends of CLD vs. mean log content of CLD, 5bCLD, and 5bCLD / CLD ratio (from left to right – natural logarithm) in SW, according to a) soil, and b) geology (for soil and geology, see legend in th

---

## Author Response (AR1)

Manuscript Number: hess-2018-377

**A conceptual model 1 of organochlorine fate from a combined analysis of spatial and mid/long-term trends of surface and ground water contamination in tropical areas (FWI)**

by Philippe CATTAN, Jean-Baptiste CHARLIER, Florence CLOSTRE, Philippe LETOURMY, Luc ARNAUD, Julie GRESSER, Magalie JANNOYER

We thank S. Reichenberger and the second reviewer for their helpful comments provided in the review file hess-2018-377-RC1 and hess-2018-377-RC2. We detail in this file the response to each comment and the corresponding changes that we propose to improve the manuscript:

- each comment of the referee about the original version (OV) of the manuscript (indicated in black),
- our corresponding responses (indicated in blue)
- *our proposals for amending the text (in blue and in bold-italic),*

**Response to the first reviewer**

I think this paper is of good quality and contributes significantly to the understanding of the environmental fate of CLD on the island of Martinique. However, I have a few comments (see below) that should be addressed in order to make the paper easier to read and understand.

**General comments:**

1) The word "residence time" is somewhat vague. Please specify at first mention what is exactly meant (e.g. mean transit time).

Response: the groundwater residence time refers to the water age in aquifers defined as the mean transit time by Maloszewski and Zuber (1992). It would be addressed in the revised version of the manuscript.

Change in manuscript: L70 "Moreover, at depth, contrasting residence times of water *(the water age in aquifers is defined as the mean transit time (Małoszewski and Zuber, 1982))* in aquifers, ranging from several years to several decades, partly account for the variability in GW contamination by CLD (Gourcy et al., 2009)*"*

2) You mention in l. 432 that the 5bCLD/CLD ratio in the commercial product Curlone was 0.0011. How was the ratio in the product Kepone that was applied before? Was the 5bCLD/CLD ratio in the commercial products constant over time, or did it vary between batches of the same product?

Response: according to Devault et al. (2016), Kepone and Curlone products had very similar ratio characterized by values of 0.00077. In our text, we specified a value form Clostre et al (2015). We propose to change to Devault et al. (2016) for consistency. Notice that Devault et al. also compared three different batches of Curlone without significant difference. Equally, we modified paragraph 3.1 according to your comment 18.

Change in manuscript: L430 "This was hypothesized by observing the distribution of 5bCLD / CLD ratios in water (median of 0.03; 1st centile of 0.006) with a far higher median and first centile value than in the commercial products ***Kepone® and Curlone® used in FWI (mean ratio of 0.00077 ± 0.00027(Devault et al., 2016))***".

3) Can your findings / model be extrapolated to other CLD-contaminated areas in the Antilles, notably the island of Guadeloupe?

Response: indeed, our conceptual model could be generalized to Guadeloupe archipelago, where chlordecone was also applied intensively in banana plantations.

Change in manuscript : L441 end of § *" **These results on the island of Martinique could indeed be extrapolated to other CLD-contaminated areas, such as in the Guadeloupe archipelago (FWI) where CLD was also intensively applied in banana plantations.**"*

**Specific comments:**
Abstract:
4) p. 1, l. 28: "old geology": I know what you mean, but maybe "old geological substrates" or "old geological formations" would be more appropriate?

Response: we agree with comment and "old geology" was replaced by "***old geological formations***" throughout the text in the revised version

5) p. 1, l.29: "theoretical leaching model": maybe "conceptual leaching model" (as in the manuscript title)

Responses: We agree. Change OK

**Introduction:**
6) p. 2, l. 64-65: "acute" and "environmental" are not opposites; better use "chronic" instead of "environmental" (exposure via the environment can be acute or chronic)

Response: We agree. Change OK

7) p. 2, l. 69: "partitioning coefficient (Koc) between the sorbed part on soil organic matter": not comprehensible ! needs to be rephrased

Response: OK. We propose to rephrase as follows, just giving the name of the coefficient:

Change in manuscript:L69  "***soil organic carbon-water partitioning coefficient*** (Koc)"

8) p. 2, l. 71: "contrasting residence times": What residence times: of water or of CLD?

Responses: it refers to residence time of water and it will be addressed in the revised version.

Change in manuscript  L70 See our response to comment 1) where we propose to add the following sentence: *"...constrasting residence times **(the water age in aquifers defined as the mean transit time (Małoszewski and Zuber, 1982))** in aquifers of…"*

**Materials and Methods:**

9) p. 3, l. 111-112: "ferralitic soils (latosols) ! ferralsols": What is the difference between the two? The WRB system contains the reference soil groups ferralsols and plinthosols, but not "ferralitic soil (latosols)".

Response: It is a mistake. The climate sequence described in Comet-Daage et al. (1965) is latosols -> ferrisols -> vertisols. A correspondence with the WRB system is given by Delvaux and Brahy (2014) and is *ferralsol -> nitisol -> vertisol*. We corrected.

Change in manuscript : L111-112 "*ferralsol -> nitisol* -> vertisol"

10) p. 6, l. 213: "measurable": maybe more precisely quantifiable", since it refers to the LOQ

Response: We agree. Change OK

11) p. 6, l. 215: "data item" (or data point): How is this defined? Unique combination of water sample and compound?

Response and Change in manuscript: OK. We propose to specify in the text "data item (i.e. *water samples analyzed for CLD and 5bCLD)*"

12) p. 6, l. 230: eq. 1: explain the indices (i, j, k, l, m, t)

Response: There is a mistake in the text and in Equations 1 and 2. We corrected and specified in the text:

Change in manuscript: L 233 end of § "*. Indices i, j, t, k, l represent following factors soil x geology, hydrological sector, date, sampling point and sample replication, respectively*"

13) p. 6, l. 234: "totally correlated": express more precisely

Response: because of a limited number of sites for groundwaters, there is not a so high spatial variability of geology and soil as observed for surface water data set. Thus, for groundwater data set (model 2), geology and soil are correlated: Andosol on recent geological formations, and ferralsols on old geological formations. To be clearer in the revised version, we modified the sentence as follows:

Change in manuscript: L234 "Soil and geological factors were *closely linked* for the GW data set *(andosols were always associated with recent geological formations and ferralsols with old geological formations)…*"

14) p. 6, l. 241: "dispersion indices": How can this quantity be interpreted? and 15) p. 6, l. 242: "confidence coefficient": What is this?

Response: dispersion indices can be interpreted like standard deviation. From the log values ln(x), we determined a confidence interval [lninf, lnsup] (with lninf = ln(x)- sd, lnsup = ln(x)+sd, sd the standard deviation of log value). That is real values are included in the interval [exp(lninf), exp(lnsup)]. Because exp(sd ) is not convenient, we defined a new dispersion index: [exp(lnsup)-exp-ln(inf)]/2, i.e. "half the difference between the limits of the confidence interval".

Change in manuscript: In fact, since we no more used these dispersion indices in the revised version of the article, we propose to suppress L241 and 242.

16) p. 6, l. 248: "Sen trends": What are Sen trends, and what do they mean statistically? (Explain in 1 or 2 sentences.)
Response: OK. We propose the following change

Change in manuscript: L248 "We calculated Sen trends *(Sen's slope estimator, (Gilbert, 1987)* for each variable (CLD, 5bCLD and ratio) in order to compare dynamics for the two compounds. *The Sen trend of a set of two-dimensional points (xi,yi) and (xj,yj) is the median of the slopes (yj − yi)/(xj − xi) determined by all pairs of sample points. The Sen trend is more robust than the least-squares estimator, because it is much less sensitive to outliers"*

17) p. 8, l. 258-262: What are the dimension and unit of the lixiviation rates TCLD and T5bCLD? This does not become entirely clear from eq.6 because of the various unit conversion factors. I end up with the unit 1/year.
Responses: Tcld and T5bcld are mass ratio. We propose to add a comment on that point in the text:

Change in manuscript: L258 "TCLD and T5bCLD are the rates of lixiviation for CLD and 5bCLD *(i.e. the ratio of lixiviated mass of CLD or 5bCLD to their respective mass in soil)…"*

Additionally we propose to modify Eq. (6) for consistency with regard to the dimensions
"Eq. (6):                         $T_{CLD} = (R{\times}S)/(Koc{\times}(C/1000){\times}D{\times}(d{\times}S))$
where Koc (L kg$^{-1}$) is the partitioning coefficient between the sorbed part on soil organic matter and the dissolved part in water, D (kg dm$^{-3}$) the bulk density, C (g kg$^{-1}$) the soil carbon content, and R (dm) the annual amount of rainfall, *S the soil surface area (dm²) and d the soil depth (dm)."*

**Results:**
18) p. 8, l. 279-281: It should be mentioned here how high the ratio 5bCLD/CLD was in the commercial products that were applied, and whether it was constant over time.

Response: We propose to add the following text at the end of the paragraph:

Change in manuscript: L281 end of § *"According to Devault et al. (Devault et al., 2016), these differences cannot stem from the use of different commercial products or different batches of the same product. Indeed, these authors, found no significant statistical difference between the ratio of the commercial products Kepone® and Curlone® used in FWI, no more than they did between samples from different batches of Curlone®. They found a mean ratio in commercial products of 0.00077 ± 0.00027, i.e. ten times lower than our observations in river."*

19) p. 9, l. 324-325: "shorter residence times were observed for more recent formations": (Are you referring to GW only or also to SW?) This is interesting. I would rather have expected the opposite. Can you briefly explain why hydraulic residence times (mean transit times?) are shorter in the younger geological formations of Martinique than in the older ones?

Response1: We refer to groundwaters (knowing that groundwaters are the main contributor of contamination of surface waters). We propose to add the following text in the 2.1 Section:

Change in manuscript: L134 after "old weathered formations". "*Given that the weathering of geological formations increases with their age, it is the main cause of a global decrease in aquifer permeability, notably in volcanic regions (Lachassagne et al., 2014). Indeed, clayey alteration products by weathering constrain the soil's physical and hydrodynamic properties by reducing porosity, and consequently permeability (Adelinet et al., 2008)*"

Response2: Thus, as we observe higher 5b/CLD ratio on younger geological formations (i.e. unweathered formations), we hypothesis that this was related to a shorter residence time. We propose to modify the sentence as follows: L324

Change in manuscript: L324 "Thereby, shorter residence times were observed for *aquifers located in* more recent *and unweathered geological* formations"

20) p. 10, l. 344: "water CLD contents below the detection limit appeared less frequently": meaning not entirely clear ! rephrase

Response: We propose the following text:

Change in manuscript: L343 "For the two sites showing a decrease in water CLD content, *the number of samples with* 5bCLD contents below the detection limit *decreased over time*, and *equalled zero* in the case of the Source Morne Figue site after 2011"

21) p. 10, l. 384: unit for bulk density is missing

Response: We corrected "*kg dm$^{-3}$*"

22) p. 13, l. 457: "unweathered formations favour rapid transfers"; Why is that? (cf. comment on l. 324-325)

Response: please see our response to comment "19)"

23) p. 13, p. 468: "we cannot assess it": Assess what?

Response: the effect of soil on degradation process. We modified the text accordingly.

**Conclusions:**
24) p. 13, 507-508: "This led to implications regarding where and how to act to reduce impacts": Can you elaborate on this further? Is there really anything that can be done except waiting for CLD to degrade and leach from the system?

Response: We propose some examples:

Change in manuscript: L508 "*(e.g. choice of crops according to pollution levels since some plants are less sensitive to contamination than others (Clostre et al., 2015), constraints on*

*water management like drinking water and irrigation, choice of priority areas to test decontamination processes, setting up compensation plans according to the risk…)"*

**Figures:**
25) Figure 3: The figure is neat, but too small for reading the legend or for identifying much on the map. ! upscale

Response: We propose a new version of Figure 3

Change in manuscript: change of Figure 3 and relative caption: *"Distribution of water CLD content (a, c, e) and 5bCLD / CLD ratio (b, d, f) for surface water (square) and groundwater (star), according to banana cultivated areas and hydrological sectors (a and b), soils (c and d) adapted from Colmet Daage (1965), and geology (e and f) adapted from Germa et al. (2011).* Large squares are relative to sample points having more than ten sampling dates and small squares having fewer than ten sampling dates"

26) Figure 4: y-axis: The numbers are difficult to interpret. Try lg or non-logarithmised numbers instead of ln.

Response: As specified in material and methods, data were log transformed for all analysis. For Figure 4 we worked with log-transformed data. We propose to complete the caption Figure 4 specifying the correspondence between log and non-log values:

Change in manuscript: *"The y values of -6, -4 and -2 correspond to ratio values of 0.002, 0.018 and 0.135, respectively."*

27) Figure 4: give the unit of formation age (million years) directly in the figure (e.g. 1.0-0.3 Ma)

Response: This is done

28) Figure 6: too small, needs upscaling (if it takes too much space after upscaling, consider shifting it to a SI section).

Response: We propose a new Figure

29) Figure 7: If one doesn't know what Sen trends are, the figure is not understandable.

Response: This is now explain in the text accordingly to the response to your comment N°16

**Tables:**
30) Table 2: Table header needs to be rephrased to improve understandability.

Response: Caption and headers were modified in the revised version. We propose the following headers that matches terms in the caption and the text:

Change in manuscript: *Simulation , target value, fixed parameter*

Best regards,

Stefan Reichenberger

References:

*Adelinet M., J. Fortin, N. d'Ozouville, S. Violette, 2008. The relationship between hydrodynamic properties and weathering of soils derived from volcanic rocks, Galapagos Islands (Ecuador). Environ Geol (2008) 56:45–58, DOI 10.1007/s00254-007-1138-3.*

*Clostre, Florence, Philippe Cattan, Jean-Marie Gaude, Céline Carles, Philippe Letourmy, and Magalie Lesueur-Jannoyer. 2015. "Comparative Fate of an Organochlorine, Chlordecone, and a Related Compound, Chlordecone-5b-Hydro, in Soils and Plants." Science of The Total Environment 532 (November): 292–300.* https://doi.org/10.1016/j.scitotenv.2015.06.026.

*Delvaux, B.; Brahy, V. Mineral Soils conditioned by a Wet (Sub)Tropical Climate."FAO. Retrieved 14 June 2014.*

*Devault, D. A., Laplanche, C., Pascaline, H., Bristeau, S., Mouvet, C. and Macarie, H. 2016. Natural transformation of chlordecone into 5b-hydrochlordecone in French West Indies soils: statistical evidence for investigating long term persistence of organic pollutants, Environ. Sci. Pollut. Res., 23(1), 81–97, doi:10.1007/s11356-015-4865-0.*

*Gilbert, Richard O. 1987. Statistical Methods for Environmental Pollution Monitoring. New York: Wiley.*

*Lachassagne P., B. Aunay,N. Frissant, M. Guilbert, and A. Malard, 2014. High-resolution conceptual hydrogeological model of complex basaltic volcanic islands: a Mayotte, Comoros, case study. Terra Nova, 26, 307–321, doi: 10.1111/ter.12102.*

*Maloszewski, P., Zuber, A., 1982. Determining the turnover time of groundwater systems with the aid of environmental tracers: I. : Models and their applicability, J. Hydrol., 57, 207-231.*

**Response to the second reviewer**

…key conceptual assumptions have to be thoroughly discussed and significantly improved to consider publication in HESS.

First, using a transformation product (TPs) 5bCLD as a tracer of degradation extent and associated persistence is a valuable idea, but the degradation of the TP itself is never discussed. Similarly the model seems to consider as a perfect tracer 5bCLD, i.e. without degradation. This main assumption can significantly alter the assessment of persistence done and this point is never discussed.

Response: We thank the reviewer for raising this important point. Degradation of the TP can be discussed adding a new calculation step in the model accounting for 5bCLD degradation. Eq (4) can be modified as follow:

$$5bCLD(t + 1) = 5bCLD(t) - 5bCLD(t) \times T_{5bCLD} - 5bCLD(t) \times C_{5bdegrad} + CLD(t) \times C_{degrad}$$

[Figure]

Then, it is possible to test 3 values of $C_{5bdegrad}$ in a wide range surrounding the one of CLD ($C_{degrad}$): $C_{5bdegrad} = 0$, $C_{5bdegrad} = C_{degrad}$, $C_{5bdegrad} = 10 \times C_{degrad}$. Notice these values are highly speculative since there is no experimental $C_{5bdegrad}$ value and that $C_{degrad}$ is the result of optimization process in our paper. Results are reported in the Figure above which shows the evolution of the 5bCLD lixiviation and of the ratio for the 3 tested values. The figure shows similar dynamics of ratio evolution or of lixiviation evolution. The difference between the simulations remains weak, notably because the tested values are about 10 and 100 times lower than lixiviation rate ($T_{5bCLD}$ equals 0.1242 here while $C_{degrad}$ equals 0.0014)). Consequently, introducing a degradation coefficient does not alter here our first conclusions. Running optimization process with this new term, we find $T_{5bCLD} = 0.1242$, $C_{degrad} = 0.0014$ and $C_{5bdegrad} = 0.0010$.

Our assumptions are also consistent with estimations of Dolfing et al. (2012) showing that the solubility is higher for transformation products of CLD.

Change in manuscript:
So, to account for the reviewer comment, we propose
- 1) To complete the current model adding a degradation term for 5bCLD.
- 2) Given the lack of knowledge and the uncertainty about degradation rate, we propose to add the following comment in the text L390 *"It should be noticed that the degradation values remained uncertain as we did not have any references for comparison. In our case, the optimization process yielded a far lower degradation rate compared to the lixiviation rate (Table 2). Consequently, the model will be less sensitive to changes in the degradation rate than in the lixiviation rate, which is the key parameter for determining the ratio in water. Additionally, there was uncertainty when comparing degradation rates for 5bCLD and CLD. The optimization process yielded degradation rates for 5bCLD and CLD of the same order of magnitude. Additional simulations showed that setting C5bdegrad ten times higher than Cdegrad instead of zero reduced the 5bCLD / CLD ratio by 10 percent without changing the dynamic of the ratio and of 5bCLD lixiviation (not shown). Given that CLD transformation products are likely to be more mobile in the environment than their parent compound (Dolfing et al. 2012), we assumed that our model gave*

*sufficient bases for interpreting our results."*.

Do you have access to CLD/5bCLD ratio in soil to have an idea of the initial signature over time to decipher soil degradation process to those associated to surface and groundwater flowpaths?

Response: Reference of ratio in soils are in the paper of Clostre et al (2015).The median value of 0.011 in nitisols and 0.017 in andosols were used in our paper to constrain our model (see section 3.4). This does not help to speculate about ratios in water since they depend on lixiviation rates of CLD and 5bCLD. In our article, data from Cabidoche et al (2009) were used to assess CLD lixiviation rate ($T_{CLD}$) for andosols and nitisols. The 5bCLD lixiviation rate ($T_{5bCLD}$) stemming from the optimization process appears higher than $T_{CLD}$. This result is consistent with Devault et al (2016) who conclude for a higher mobility for 5bCLD than for CLD. Whatever, it is unlikely that CLD was leached while 5bCLD accumulated in soil profile due to the highest mobility of transformation products (Dolfing et al., 2012).

Change in manuscript: We propose to add the following sentence section 3.4 L406: "… continuously without a plateau. ***This result was consistent with Devault et al (2016) who concluded on higher mobility of 5bCLD compared to CLD, and more generally with the results of (Dolfing et al., 2012), who showed that transformation products had higher mobility than CLD.***"

Second, residence time is used to explain the spatial variability of the ratio compounds/TPs. To support the discussion, the authors should provide existing reported information/simulation of these residence times:
- to discuss spatially contrasted compounds/TPs ratio delivery by soil to ground water

Response: please see our response to the comment "19)" of the First referee

- to address the question of degradation of the TP itself (especially for long residence time)

Response: longer residence time does not mean that the TP degradation is higher. In fact degradation occurs in the soil, whereas residence time in the aquifer refers to transfers in depth (below soil cover, through the unsaturated and saturated zones) where the degradation (as well as the retention) is considered as null. Groundwater residence time is generally superior to several years (up to several decades – see Gourcy et al., 2009 for instance) that is widely superior to the residence time of the infiltrated water in the soil cover (several days or months).

Third, if the sampling effort, statistical analyses and conceptual development provided a coherent approach for groundwater (slow flowpath), I have many questions on the surface water component.

Response: Global comment about flowpath, as a state of the art for our following responses to the several comments related to that aspect. This global comment justifies our approach and shows in what way the integration of the surface water component will not affect our conclusions.
First, volcanic soils in Caribbean islands have a high infiltration capacity (saturated hydraulic conductivity superior to 60 mm/h (Cattan et al., 2006; Crabit et al, 2016). Then, despite high rainfall intensities and amounts, most of rainfall infiltrates (about 95% at the plot scale according to Cabidoche et al, (2009); more than 90% at the watershed scale according to Charlier et al., 2008; 2011) generating either subsurface or deep flows. So leaching is the main process in pesticide transport in surface water.

Second, usually, one reason to study separately pesticide transport by surface runoff is that the pesticide concentration in runoff water may vary highly according to time of pesticide application at the plot scale (Saison et al., 2008) as well as at the watershed scale (Charlier et al., 2009). It is not the case for CLD which have been applied long time ago: boundary conditions relative to pesticide concentration in soil are almost steady. Surely, during application period, agricultural practices may have affect 5bCLD/CLD ratio day by day. However our model aims to simulate the ratio evolution over a long time period. A second reason to consider separately runoff and infiltration water is that pesticide concentration in surface water at the plot scale may differ from infiltrated water. There are few references about this point for CLD. Cabidoche et al (2009) notice that CLD concentration in surface runoff was more than 3-fold lower than in drainage, while runoff volume was 10 times lower than drainage volume. They consequently neglected loads in runoff that represented less than 1/30 of those in drainage at the plot scale.

Given the previous consideration, we then choose to focus here on lixiviation process, which affect the ratio dynamic on the long term. The reviewer ask the question of the effect of event-driven process (storm event, surface runoff, erosion, application practices) on long term trends and how they can modify CLD concentration in water and the ratio. It is a difficult issue that would require getting spatial distribution of storm event, and their contribution to river pollution. This lack of knowledge probably leads to minor CLD exportation. Indeed, most of the time (even in rainy regions), surface flow in the river is driven by baseflow from aquifer's drainage, originated from water infiltration. Knowing that groundwater concentrations are widely higher than in rivers, concentrations during storm events would lead to generate diluted concentrations in surface waters.

We propose different changes relatively to the reviewer comments. We equally propose to add a § "main assumption about CLD transfer" in discussion section

Change in the manuscript: L417 addition of the § "*4.1 main assumptions about CLD transfer.*
*In our study, we focused on long-term trends for CLD and 5bCLD concentration in water, along with their ratio. We considered that the main process determining pollutant concentrations in water was relative to CLD desorption by water infiltrating the soil. We assumed this hypothesis for different reasons.*
*Firstly, rain water mainly infiltrates. In fact, given the high soil infiltration rate (saturated hydraulic conductivity over 60 mm/h (Cattan et al., 2006; Crabit et al., 2016), most rainfall infiltrates (about 95% on a plot scale according to Cabidoche et al, (2009); more than 90% on a watershed scale according to Charlier et al. (Charlier et al., 2008, 2011)), generating either subsurface or deep flows. Consequently, transportation by surface runoff is low. Cabidoche et al (2009) found that CLD concentration in surface runoff was more than 3-fold lower than in drainage, while the runoff volume was 10 times lower than the drainage volume. They consequently discarded loads in surface runoff that amounted to less than 1/30 of those in drainage on a plot scale*
*Secondly; soils have little erodibility: Cabidoche et al (2009) found that "All the soil types in FWI are acidic, which prevents clay dispersion and sheet erosion. Hydric erosion*

*appears to be due only to bad soil management practices, which concentrate runoff that then forms streams that are able to carry aggregates". Thus, erosion from cultivated soils is probably not a major way of CLD transportation. Moreover, given the high contribution of erosion from river beds and from non-contaminated areas in the upstream zone (due mostly to torrential type flow of rivers in FWI), the impact of surface water contamination by sediments was considered as a minor process.*
*Lastly, by neglecting transport via surface runoff (since sampling mainly occurred outside storm event periods), we probably underestimated pollutant exportation. Thus, we expected that it should not have a great impact on the long-term dynamics of concentrations and ratios in rivers, which is one of the main topics of our paper.*

- The representativeness of the sampling (low frequency mainly during based-flow, if I well understood the database characteristics) is not discussed taking into account percentage of Chlordecon exported during storm event associated to tropical climat.

Response: See the last paragraph of the new 4.1 Section

Change in manuscript: ….*"Finally, neglecting transport via surface runoff (since sampling mainly occurred outside storm event periods), we probably underestimated pollutant exportation"*…

- With a large Koc, the question of Chlordecon released from soil to river by erosion during runoff event is never discussed. How these pulses can contribute to spatial and temporal patterns of chlordecon in surface water?

Response: See the following sentences in the new 4.1 Section

Change in manuscript: *" Second soils have little erodibility: Cabidoche et al (2009) notice that "All the soil types in FWI are acidic, which prevents clay dispersion and sheet erosion. Hydric erosion appears to be due only to bad soil management practices, which concentrate runoff that then forms streams that are able to carry aggregates".Thus, erosion from cultivated soils is probably not a major way of CLD transportation. Moreover, given the high contribution of erosion from river beds and from non-contaminated areas in the upstream zone (due mostly to torrential type flow rivers in FWI), the impact of surface water contamination by sediments is considered as a minor process."*

- For surface water, it could be relevant to know if the CLD concentrations correspond only to the dissolved phase or if it is a "total" concentration.

Response: the CLD concentration is a total concentration. See the change we suggest for the following comment L153

- Information on the filtration and purification steps are not provided in the M&M section.

Response: there was no purification nor filtration since the suspended matter content of samples was low (less than 250 mg L$^{-1}$). Analyses were performed on raw water. We propose to add the following sentences section 2.2.2

Change in manuscript: L153 *"Analyses were carried out on raw sampling water. Thus, the water CLD and 5bCLD contents corresponded to dissolved and particulate fractions. It should be noted that the particulate fraction of the samples was low (< 250 mg L-1) due to sampling conducted mainly during periods of low flow."*

- Can contaminated sediments in river potentially be remobilized by event and alter trend assessment in surface water?

Response: see our previous response

- In the conceptual model, the surface runoff and the surface water to groundwater seem not considered. The choice targeting mainly leaching and not the other off-site transport is never discussed. The authors mentioned "hope for pollution mitigation" based on statistical model, but I wonder how fast flow in river can modify this assessment.

Response: see our previous responses

In my opinion, the paper can't be published without strengthening of these points.

Specific remarks:
L324 GW, as well as in SW fed by it. And vice et versa ?

Response: whereas infiltration from ditches towards aquifers is a likely process in such regions due to the high permeability of the shallow formations (Charlier, 2007), and even if in some cases, river infiltrations may contribute also to groundwater recharge (Charlier et al., 2011), we consider that the infiltration of surface water is a neglecting process of groundwater contamination at a global scale. In fact, in cultivated areas, surface water is generally widely less contaminated in CLD than groundwaters.

L323 The age of the main geological units was used as an indicator of hydrogeology and notably residence time in the aquifers. Could you provide evidence ? Residence time assessment form others studies ?

Response: please see our response to the comment "19)" of the first referee

L57 and L60, 1993 or 1992, I guess banned in 1992 but used until 1993. Please explain.

Response: Yes there was exemption until 1993. We propose:

Changes in the manuscript: L60 "...ban in 1992 *(there was exemption in FWI until 1993)"*

L121 "they are intergrades" ?

Response: Intergrades are defined by Colmet-Daage relative to the climatic sequence ferralsols -> vertisols for soils that are "intermediate". Since Colmet-Daage classification is specific, we propose to suppress the last part of the sentence which is unclear ""

L134 unweathered formations, to several decades for old weathered formations (provide range for "Old")

Change in manuscript: *"between a few years for recent unweathered formations **(<0.5-1My)**, to several decades for old weathered formations **(> 1My)**"*

L139 routine basis with CLD. For (double space before For)

Response: OK

L150 5bCLD is the main CLD co- and alteration product of CLD: what do you mean by alteration product ? Transformation / degradation product ? Please clarify ?

Response: In fact, 5bCLD can be considered both as a co-product and as a degradation product. Consulting biochemists, the word "alteration" seemed more convenient. We propose the following change:

Change in manuscript: *"**5bCLD is the main alteration product of CLD (the term "alteration" here means that 5b is both a co-product and a degradation product)** for which …"*

L151 "Reference standards for CLD and 5bCLD were purchased" : provide purity degree

Change in manuscript: L152 *"**…** for both laboratories **with a purity degree of 96.7%.**"*

L149 Between sampling and analysis, no information is provided on the filtration (raw water/filtrated water?) , purification ?, please add

Response: OK analyses were performed on raw water. See previous change for L153

L181 was not detected (i.e. 0.001 for LDA26 or 0.003 µg L−1 for BRGM), and an intermediate value of 0.006 µg.L−1 : why is different of value provided in L177 0.003 µg.L−1 ? Please clarify

Response: 0.003 line 177 refers to the limit of detection; 0.006 is an intermediate value between the limit of detection 0.003 and the limit of quantification 0.01 when the compound was detected but not measurable. We propose to change "measurable" by "*quantifiable*" L182

L184 double space the 5bCLD

Response: OK

L183 Factors. Not clear for me, how heterogeneity of upstream catchment for SW or drainage area for GW were integrated in metrics ?

Response: factors refer to global descriptors that do not integrate such spatial heterogeneity at a local scale. Apart soil (as it is explained in the text), each site is associated with the factor value at the sampling point.

L204. For GW, double space

Response: OK

Fig. 1. I suggest to modify this figure to add sampling point distribution (the different zoom levels can be significantly reduced)

Response: sampling distribution are presented in figure 3

L248 Kendall (MK) test. We calculated Sen trends, Sen trend ? Not defined, Instead to use Sen trend in the text, I suggest to explain the information underlined by this metric (to improve understanding for the reader)

Response: OK, please see our response to the comment 16) of the first Referee

The section 3.2.1. looks like a figure caption (modify and interpret directly in this section)

Response: Section 3.2.1 aims to present Figure 3 and the distribution of pollution

L300 "areas since 1970, i.e. during CLD application. Surprisingly, SW and GW contamination occurred outside these banana areas" Explanation ? other dissipation Processes ? Are the contaminated areas are downstream of banana areas ?

Response: we suggested CLD misuse L304

Fig3. Legend cannot be read (too small).

Response: We propose a new Figure with a larger legend

Fig4. two small, I suggest to merge some of them or provide in SI

Response: there are two comments for Figure 4. Perhaps this comment is relative to Fig3 ? or Fig6. We propose a new Figure 6

L308 contamination level. For example, the CLD content of hydrographic subsector 1 (see Figure 3 left for locations) was different from hydrographic subsector 2 even though the points in each zone had the same contamination level. It is very descriptive, please provide explanation

Response: We propose to rephrase L308-309

Change in manuscript: *"For example, although sample points of subsector 1 and 2 were very close (see Figure 3a), they did not have the same contamination level. In contrast, all the sample points of subsector 1 had the same contamination level (same for subsector 2). This suggests that the hydrographic sector, i.e. the water flows within the same hydrological unit, mainly determined the contamination level of the sample points, rather than the geographical closeness of those points."*

L 320 "This statistically confirmed"/ Missing word ?

Response and change L320: We propose to rephrase: *"This is statistical confirmation of the result mapped in Figure 3…"*

Figure 4. provide full name under the figure Ferr_And, Nit_And. . .

Response: OK

L375 : "duration of pollution": persistance of pollution ?

Change in manuscript: L375 *"persistence of pollution"*

Fig 7. Legend is hidding some point: modify. I suggest to redesign the figure 7 to improve understanding of key message for the reader (add sectors/types and assessment indicator) ???

Response: We propose to keep the legend of the figures in the middle and to suppress the legend of figures on the left and right sides where points are hidden. Sen trends for others factors (hydrographic sectors and historical banana area) are not represented due to the absence of relationships.

Change in manuscript: redesign of figure 7 whith and a new caption: "Sen trends of CLD vs. mean log content of CLD, 5bCLD, and 5bCLD / CLD ratio (from left to right – natural logarithm) in SW, according to a) soil, and b) geology *(for soil and geology, see legend in the middle figure)*.

Fig 8. time unit ? Years. . . As discussed in the main comments, all the model predictions seem to be dependant of persistence of the 5bCLD, how the results could be altered by considering TPs degradation.

Response: see response above

L437 0.1 µg L-1 437 during baseflow periods (flood flow periods being rarely sampled) given a current concentration of 0.5 µg L-1 438 on average. I don't understand your assumption?

Response: baseflow periods refers to periods without flood flows (or storm flows). Please see also our response to your previous comments on sampling

L499 "catchment scale", you used watershed during all the manuscript why changed now? "The residence time - estimated by the water apparent age: not discussed or characterized before?

Response and change: catchment is replaced by watershed. Regarding the residence time, it was discussed in L452-458 of the submitted version

L388 "they should lie": sentence ?

Response and change L462: "…considering *the ratios should lie*…"

Additional references

*Cattan, P., Y.-M. Cabidoche, J.-G. Lacas, and M. Voltz. 2006. Effects of tillage and mulching on runoff under banana (Musa spp.) on a tropical Andosol. Soil Tillage Res. 86:38–51.*

*Charlier, J.-B. 2007. Fonctionnement et modélisation hydrologique d'un petit bassin versant cultivé en milieu volcanique tropical. Ph.D. diss. Université des Sciences et Techniques du Languedoc, Montpellier II.*

*Dolfing J., I. Novak, A. Archelas, and H. Macarie, 2012. Gibbs Free Energy of Formation of Chlordecone and Potential Degradation Products: Implications for Remediation Strategies and Environmental Fate. Environ. Sci. Technol., 2012, 46 (15), pp 8131–8139. DOI: 10.1021/es301165p*

*Saison, C., P. Cattan, X. Louchart, and M. Voltz. 2008. Eff ect of spatial heterogeneities of water fl uxes and application pattern on cadusafos fate on banana cultivated andosols. J. Agric. Food Chem. 56:11947–11955.*

[revised manuscript text omitted]